# SetCSE: Set Operations using Contrastive Learning of Sentence Embeddings

**Kang Liu**
Independent Researcher
`kangliu@umich.edu`

## Abstract

Taking inspiration from Set Theory, we introduce SetCSE, an innovative information retrieval framework. SetCSE employs sets to represent complex semantics and incorporates well-defined operations for structured information querying under the provided context. Within this framework, we introduce an inter-set contrastive learning objective to enhance comprehension of sentence embedding models concerning the given semantics. Furthermore, we present a suite of operations, including SetCSE intersection, difference, and operation series, that leverage sentence embeddings of the enhanced model for complex sentence retrieval tasks. Throughout this paper, we demonstrate that SetCSE adheres to the conventions of human language expressions regarding compounded semantics, provides a significant enhancement in the discriminatory capability of underlying sentence embedding models, and enables numerous information retrieval tasks involving convoluted and intricate prompts which cannot be achieved using existing querying methods.

## 1 Introduction

Recent advancements in universal sentence embedding models (Lin et al., 2017; Chen et al., 2018; Reimers & Gurevych, 2019; Feng et al., 2020; Wang & Kuo, 2020; Gao et al., 2021; Chuang et al., 2022; Zhang et al., 2022; Muennighoff, 2022; Jiang et al., 2022) have greatly improved natural language information retrieval tasks like semantic search, fuzzy querying, and question answering (Yang et al., 2019; Shao et al., 2019; Bonial et al., 2020; Esteva et al., 2020; Sen et al., 2020). Notably, these models and solutions have been primarily designed and evaluated on the basis of single-sentence queries, prompts, or instructions. However, both within the domain of linguistic studies and in everyday communication, the expression and definition of complex or intricate semantics frequently entail the use of multiple examples and sentences "collectively" (Kreidler, 1998; Harel & Rumpe, 2004; Riemer, 2010). In order to express these semantics in a natural and comprehensive way, and search information for in a straightforward manner based on the provided context, we propose Set Operations using Contrastive Learning of Sentence Embeddings (SetCSE), a novel query framework inspired by Set Theory (Cantor, 1874; Johnson-Laird, 2004). Within this framework, each set of sentences is presented to represent a semantic. The proposed inter-set contrastive learning empowers language models to better differentiate provided semantics. Furthermore, the well-defined SetCSE operations provide simple syntax to query information structurally based on those sets of sentences.

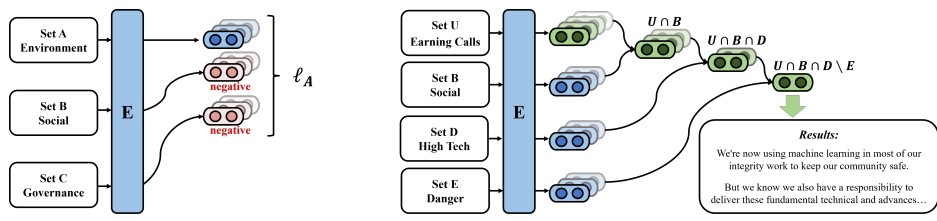

Step 1. Inter-set Contrastive Learning          Step 2. Complex Query by SetCSE Operations

Figure 1: The illustration of inter-set contrastive learning and SetCSE query framework.

As illustrated in Figure 1, SetCSE framework contains two major steps, the first is to fine-tune sentence embedding models based on inter-set contrastive learning objective, and the other is to

retrieve sentences using SetCSE operations. The inter-set contrastive learning aims to reinforce underlying models to learn contextual information and differentiate between different semantics conveyed by sets. An in-depth introduction of this novel learning objective can be found in Section 3. The SetCSE operations contain SetCSE intersection, SetCSE difference, and SetCSE operation series (as shown in Figure 1 Step 2), where the first two enable the "selection" and "deselection" of sentences based on single criteria, and the serial operations allow for extracting sentences following **complex** queries. The definitions and properties of SetCSE operations can be found in Section 4.

Besides the illustration of this framework, Figure 1 also provides an example showing how SetCSE can be leveraged to analyze S&P 500 companies stance on Environmental, Social, and Governance (ESG) issues through their public earning calls, which can play an important role in company growth forecasting (Utz, 2019; Hong et al., 2022). The concepts of ESG are hard to convey in single sentences, which creates difficulties for extracting related information using existing sentence retrieval method. However, utilizing SetCSE framework, one can easily express those concepts in sets of sentences, and find information related to "*using technology to solve social issues, while neglecting its potential negative impact*" in simple syntax. More details on this example can be found in Section 6.

This paper presents SetCSE in detail. Particularly, we highlight the major contributions as follows:

1. The employment of sets to represent complex semantics is in alignment with the intuition and conventions of human language expressions regarding compounded semantics.
2. Extensive evaluations reveal that SetCSE enhances language model semantic comprehension by approximately 30% on average.
3. Numerous real-world applications illustrate that the well-defined SetCSE framework enables complex information retrieval tasks that cannot be achieved using existing search methods.

## 2   RELATED WORK

**Set theory for word representations.** In Computational Semantics, set theory is used to model lexical semantics of words and phrases (Blackburn & Bos, 2003; Fox, 2010). An example of this is the WordNet *Synset* (Fellbaum, 1998; Bird et al., 2009), where the word *dog* is a component of the synset {*dog, domestic dog, Canis familiaris*}. Formal Semantics (Cann, 1993; Partee et al., 2012) employs sets to systematically represent linguistic expressions. Furthermore, researchers have explored the use of set-theoretic operations on word embeddings to interpret the relationships between words and enhance embedding qualities (Zhelezniak et al., 2019; Bhat et al., 2020; Dasgupta et al., 2021). More details on the aforementioned and comparison with our work are included in Appendix A.

**Sentence embedding models and contrastive learning.** The sentence embedding problem is extensively studied in the area of nautral language processing (Kiros et al., 2015; Hill et al., 2016; Conneau et al., 2017; Logeswaran & Lee, 2018; Cer et al., 2018; Reimers & Gurevych, 2019). Recent work has shown that fine-tuning pre-trained language models with contrastive learning objectives achieves state-of-the-art results without even using labeled data (Srivastava et al., 2014; Giorgi et al., 2020; Yan et al., 2021; Gao et al., 2021; Chuang et al., 2022; Zhang et al., 2022; Mai et al., 2022), where contrastive learning aims to learn meaningful representations by pulling semantically close embeddings together and pushing apart non-close ones (Hadsell et al., 2006; Chen et al., 2020).

## 3   INTER-SET CONTRASTIVE LEARNING

The learning objective within SetCSE aims to distinguish sentences from different semantics. Thus, we adopt contrastive learning framework as in Chen et al. (2020), and consider the sentences from different sets as negative pairs. Let $h_m$ and $h_n$ denote the embedding of sentences $m$ and $n$, respectively, and $\text{sim}(h_m, h_n)$ denote the cosine similarity $\frac{h_m^\top h_n}{\|h_m\| \cdot \|h_n\|}$. For $N$ number of sets, $S_i$, $i = 1, \ldots, N$, where each $S_i$ represent a semantic, the inter-set loss $\mathcal{L}_{\text{inter-set}}$ is defined as:

$$\mathcal{L}_{\text{inter-set}} = \sum_{i=1}^{N} \ell_i, \quad \text{where } \ell_i = \sum_{m \in S_i} \log \left( \sum_{n \notin S_i} e^{\text{sim}(h_m, h_n)/\tau} \right). \tag{1}$$

Specifically, $\ell_i$ denotes the inter-set loss of $S_i$ with respect to other sets, and $\tau$ is for temperature setting.

As one can see, strictly following Equation 1, the number of negative pairs will grow quadratically with $N$. In practice, we can randomly pick a subset of the combination pairs with certain size to avoid this problem.

Our evaluations find that the above learning objective can effectively fine-tune sentence embedding models to distinguish different semantics. More details on the evaluation can be found in Section 5.

## 4 SetCSE Operations

In order to define the SetCSE operations, we first quantify "semantic closeness", i.e., semantic similarity, of a sentence to a set of sentences. This closeness is measured by the similarity between the sentence embedding to the set embeddings.

**Definition 1.** The semantic similarity, $\mathrm{SIM}(x, S)$, between sentence $x$ and set of sentences $S$ is defined as:

$$\mathrm{SIM}(x, S) := \frac{1}{|S|} \sum_{k \in S} \mathrm{sim}(\mathrm{h}_x, \mathrm{h}_k), \tag{2}$$

where sentence $k$ represents sentences in $S$, and h denotes the sentence embedding.

### 4.1 Operation Definitions

For the sake of readability, we first define the calculation of series of SetCSE intersection and difference, and then derive the simpler case where only single SetCSE intersection or difference operation is involved.

**Definition 2.** For a given series of SetCSE operations $A \cap B_1 \cap \cdots \cap B_N \setminus C_1 \setminus \cdots \setminus C_M$, the result is an ordered set on $A$, denoted as $(A, \preceq)$, where the order relationship $\preceq$ is defined as

$$x \preceq y \text{ if and only if } \sum_{i=1}^{N} \mathrm{SIM}(x, B_i) - \sum_{j=1}^{M} \mathrm{SIM}(x, C_j) \leq \sum_{i=1}^{N} \mathrm{SIM}(y, B_i) - \sum_{j=1}^{M} \mathrm{SIM}(y, C_j), \tag{3}$$

for all $x$ and $y$ in $A$.

**Remark.** As one can see, the SetCSE operations $A \cap B_1 \cap \cdots \cap B_N \setminus C_1 \setminus \cdots \setminus C_M$ rank order the elements in $A$ by the similarity with sets $B_i$ and dissimilarity with $C_j$. In practice, when using SetCSE as a querying framework, one can rank the sentences in descending order and select the top ones which are semantically close to $B_i$ and different from $C_j$. Another observation is that a series of SetCSE operations is invariant to operation orders, in other words, we have $A \cap B \setminus C = A \setminus C \cap B$.

Following Definition 2, SetCSE intersection and difference are given by Lemma 1 and 2, respectively.

**Lemma 1.** The SetCSE intersection $A \cap B$ equals $(A, \preceq)$, where for all $x, y \in A$,

$$x \preceq y \quad \text{if and only if} \quad \mathrm{SIM}(x, B) \leq \mathrm{SIM}(y, B). \tag{4}$$

**Lemma 2.** The SetCSE difference $A \setminus C$ equals $(A, \preceq)$, where for all $x, y \in A$,

$$x \preceq y \quad \text{if and only if} \quad \mathrm{SIM}(x, C) \geq \mathrm{SIM}(y, C). \tag{5}$$

**Remark.** SetCSE intersection or difference does not satisfy the commutative law, in other words, $A \cap B \neq B \cap A$, and $A \setminus C \neq C \setminus A$. The advantage and limitation of the properties mentioned in Remarks are discussed in Appendix B.

### 4.2 Algorithm

Combining Section 3 and the above in Section 4, we present the complete algorithm for SetCSE operations in Algorithm 1. As one can see, the algorithm contains mainly two steps, where the first step is to fine-tune sentence embedding model by minimizing inter-set loss $\mathcal{L}_{\text{inter-set}}$, and the second one is to rank sentences using order relationship in Definition 2.

## 5 Evaluation

In this section, we present the performance evaluation of SetCSE intersection and difference. The evaluation of series of SetCSE operations are presented in details in Appendix C.4.

---

**Algorithm 1** SetCSE Operation $A \cap B_1 \cap \cdots \cap B_N \setminus C_1 \setminus \cdots \setminus C_M$

---

1: **Input:** Sets of sentences $A$, $B_1, \ldots, B_N$, $C_1, \ldots, C_M$, sentence embedding model $\phi$
2: Fine-tune model $\phi$ by minimizing $\mathcal{L}_{\text{inter-set}}$ w.r.t. $B_1, \ldots, B_N$, $C_1, \ldots, C_M$, denote it as $\phi^*$
3: **for** sentence $x$ in $A$ **do**
4:     Compute $\sum_{i=1}^{N} \text{SIM}(x, B_i) - \sum_{j=1}^{M} \text{SIM}(x, C_j)$, where all embeddings are induced by $\phi^*$
5: **end for**
6: Form $(A, \preceq)$ and rank sentences in $A$ in descending order

---

To cover a diverse range of semantics, we employee the following datasets in this section: AG News Title and Description (AGT and AGD) (Zhang et al., 2015), Financial PhraseBank (FPB) (Malo et al., 2014), Banking77 (Casanueva et al., 2020), and Facebook Multilingual Task Oriented Dataset (FMTOD) (Schuster et al., 2018).

We consider an extensive list of models for generating sentence embeddings, including encoder-only Transformer models such as BERT (Devlin et al., 2018) and RoBERTa (Liu et al., 2019b), their fine-tuned versions, such as SimCSE (Gao et al., 2021), DiffCSE (Chuang et al., 2022), and MCSE (Zhang et al., 2022), which are for sentence embedding problems, Contriever (Izacard et al., 2021), which is for information retrieval; the decoder-only SGPT-125M model (Muennighoff, 2022) is also included. In addition, conventional techniques as such TFIDF, BM25, and DPR (Karpukhin et al., 2020) are considered as well.

## 5.1 SetCSE Intersection

Suppose a labeled dataset $S$ has $N$ distinct semantics, and $S_i$ denotes the set of sentences with the $i$-th semantic. For SetCSE intersection performance evaluation, the experiment is set up as follows:

1. In each $S_i$, randomly select $n_{\text{sample}}$ of sentences, denoted as $Q_i$, and concatenate remaining sentences in all $S_i$, denoted as $U$. Regard $Q_i$'s as example sets and $U$ as the evaluation set.
2. For each semantic $i$, conduct $U \cap Q_i$ following Algorithm 1, and select the top $|S_i| - n_{\text{sample}}$ sentences. View the $i$-th semantic as the prediction of the selected sentences and evaluate accuracy and F1 against ground truth.
3. As a control group, repeat Step 2 while omitting the model fine-tuning in Algorithm 1.

Throughout this paper, each experiment is repeated 5 times to minimize effects of randomness. The hyperparameters are selected as $n_{\text{sample}} = 20$, $\tau = 0.05$, and train epoch equals 60, which are based on fine-tuning results presented in Section 7 and Appendix C.

| | | AG News-T | | AG News-D | | FPB | | Banking77 | | FMTOD | |
|---|---|---|---|---|---|---|---|---|---|---|---|
| | | Acc | F1 | Acc | F1 | Acc | F1 | Acc | F1 | Acc | F1 |
| | BM25 | 24.90 | 24.90 | 25.02 | 25.02 | 33.40 | 38.91 | 41.25 | 41.32 | 37.59 | 41.19 |
| | DPR | 25.00 | 25.00 | 25.00 | 25.00 | 33.33 | 38.81 | 41.30 | 41.35 | 38.33 | 42.87 |
| | TFIDF | 42.02 | 42.02 | 52.36 | 52.43 | 56.39 | 53.40 | 83.37 | 83.32 | 89.98 | 89.75 |
| | BERT | 43.83 | 43.74 | 52.37 | 52.05 | 55.78 | 54.21 | 53.35 | 53.34 | 80.64 | 79.59 |
| | RoBERTa | 40.75 | 40.78 | 54.58 | 54.40 | 54.69 | 53.07 | 75.10 | 64.49 | 72.60 | 70.84 |
| Existing Model | Contriever | 48.95 | 48.63 | 58.85 | 58.08 | 54.41 | 51.67 | 59.63 | 59.81 | 75.10 | 75.09 |
| Set Intersect | SGPT | 34.88 | 34.90 | 34.97 | 35.02 | 52.83 | 52.89 | 37.35 | 37.44 | 79.80 | 78.86 |
| | SimCSE-BERT | 55.28 | 55.09 | 68.07 | 67.38 | 56.13 | 53.79 | 82.69 | 82.60 | 90.01 | 89.93 |
| | SimCSE-RoBERTa | 49.68 | 49.72 | 60.76 | 60.64 | 66.11 | 64.87 | 84.90 | 84.81 | 93.43 | 93.47 |
| | DiffCSE-BERT | 49.94 | 49.95 | 61.64 | 61.31 | 50.88 | 47.78 | 83.02 | 82.87 | 91.61 | 91.42 |
| | DiffCSE-RoBERTa | 46.29 | 46.46 | 46.61 | 46.65 | 61.71 | 60.05 | 87.31 | 87.22 | 83.06 | 82.95 |
| | MCSE-BERT | 49.98 | 49.91 | 68.79 | 68.14 | 54.01 | 50.89 | 77.35 | 77.21 | 93.56 | 92.49 |
| | MCSE-RoBERTa | 46.32 | 46.29 | 57.10 | 56.88 | 55.96 | 53.32 | 85.80 | 85.69 | 94.39 | 94.30 |
| | BERT | 70.47 | 70.32 | 87.24 | 87.19 | 71.65 | 71.01 | 95.06 | 95.06 | 98.04 | 98.04 |
| | RoBERTa | 75.87 | 75.71 | 88.30 | 88.26 | 73.76 | 73.09 | 83.59 | 83.46 | 99.39 | 99.39 |
| | Contriever | 72.88 | 72.77 | 83.97 | 83.99 | 67.83 | 67.59 | 94.20 | 94.22 | 97.03 | 97.05 |
| SetCSE | SGPT | 36.64 | 36.63 | 36.01 | 36.02 | 54.13 | 54.73 | 41.88 | 41.93 | 86.94 | 86.65 |
| Intersect | SimCSE-BERT | 77.24 | 77.22 | **89.48** | **89.46** | 83.59 | 83.44 | 97.84 | 97.84 | **99.63** | **99.63** |
| | SimCSE-RoBERTa | **79.56** | **79.57** | **89.97** | **89.97** | **85.48** | **85.25** | 98.33 | 98.33 | 99.44 | 99.44 |
| | DiffCSE-BERT | 76.43 | 76.45 | 78.31 | 78.30 | 80.93 | 80.84 | 98.24 | 98.25 | **99.79** | **99.79** |
| | DiffCSE-RoBERTa | 78.02 | 78.04 | 88.63 | 88.62 | 83.89 | 83.75 | **98.49** | **98.49** | 97.89 | 97.87 |
| | MCSE-BERT | 75.04 | 63.96 | 88.77 | 88.75 | 84.03 | 83.97 | 97.76 | 97.76 | 99.53 | 99.53 |
| | MCSE-RoBERTa | **78.18** | **78.21** | 89.23 | 89.22 | **86.21** | **86.08** | **98.65** | **98.65** | 98.66 | 98.65 |
| | Ave. Improvement | **56%** | **57%** | **46%** | **47%** | **43%** | **50%** | **27%** | **28%** | **12%** | **12%** |

Table 1: Evaluation results for SetCSE intersection. As illustrated, the average improvements on accuracy and F1 are 39% and 37%, respectively.

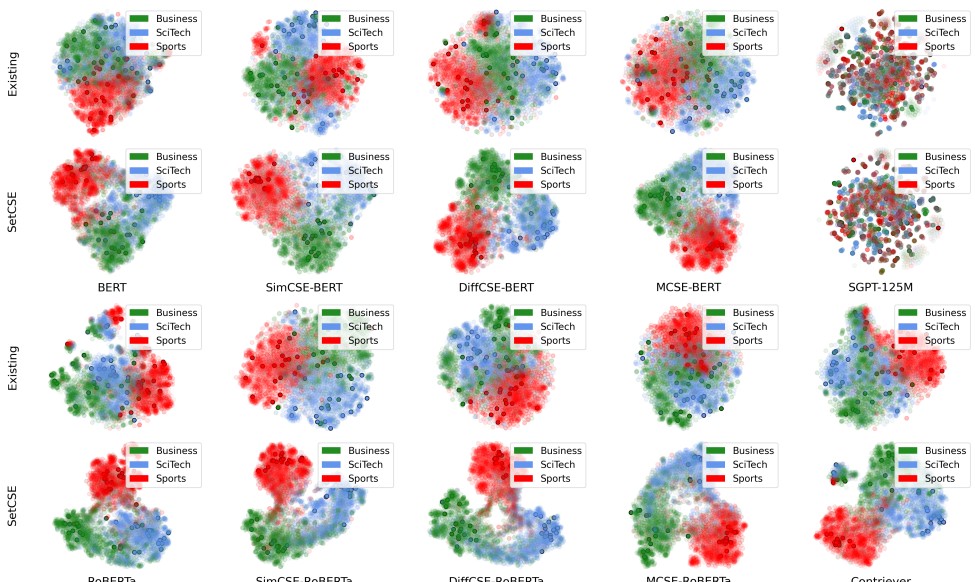

Figure 2: The t-SNE plots of sentence embeddings induced by existing language models and the SetCSE fine-tuned ones for AGT dataset. As illustrated, the model awareness of different semantics are significantly improved.

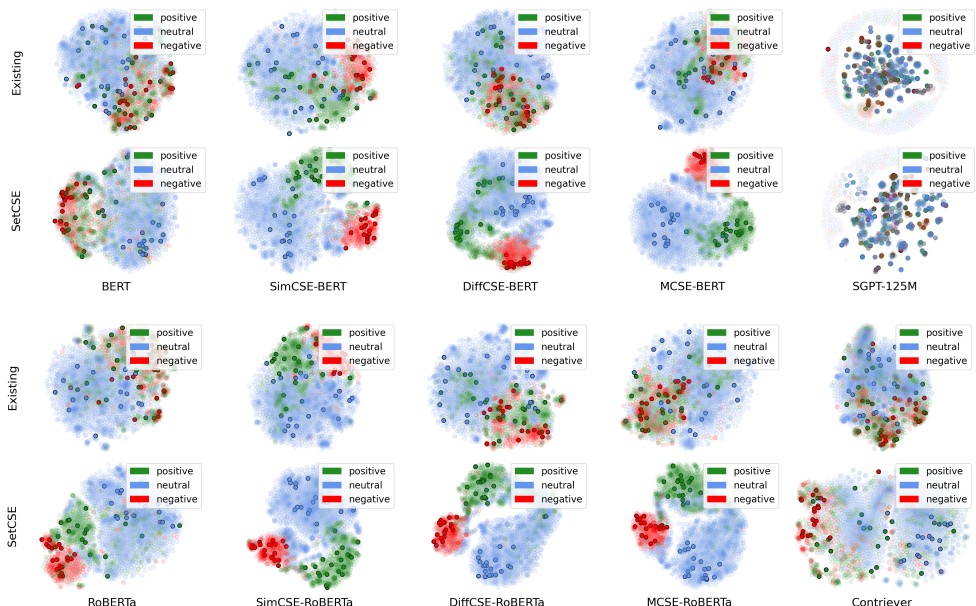

Figure 3: The t-SNE plots of sentence embeddings induced by existing language models and the SetCSE fine-tuned ones for FPB dataset. As illustrated, the model awareness of different semantics are significantly improved.

The detailed experiment results can be found in Table 1, where "SetCSE Intersection" and "Existing Model Set Intersection" represent results in Step 2 and 3, respectively. To illustrate the performance in a more intuitive manner, we include the t-SNE (Van der Maaten & Hinton, 2008) plots of the sentence embeddings, as shown in Figures 2 and 3 (refer to Section C for t-SNE plots of AGD, Banking77 and FMTOD datasets). As one can see, on average, the SetCSE framework improves performance of intersection by 38%, indicating a significant increase on semantic awareness. Moreover, the encoder-based models perform better than the decoder-based SGPT. This phenomenon and potential future works are discussed in detail in Appendix C.3.

## 5.2 SetCSE Difference

Suppose a labeled dataset $S$ has $N$ distinct semantics, and $S_i$ denotes the set of sentences with the $i$-th semantic. Similar to Section 5.1, the evaluation on SetCSE difference is set up as follows:

1. In each $S_i$, randomly select $n_{\text{sample}}$ of sentences, denoted by $Q_i$, and concatenate remaining sentences in all $S_i$, denoted as $U$.

2. For each semantic $i$, conduct $U \setminus Q_i$ following Algorithm 1, and select the top $\sum_{j \neq i}(|S_j| - n_{\text{sample}})$ sentences, which are supposed have different semantics than $i$. Label the selected sentences as "not $i$", relabel ground truth semantics other than $i$ to "not $i$" as well. Evaluate prediction accuracy and F1 against relabeled ground truth.

3. As a control group, repeat Step 2 while omitting the model fine-tuning in Algorithm 1.

Results for above can be found in Table 2. Similar to Section 5.1, we also observed significant accuracy and F1 improvements. Specifically, the average improvement across all experiments is 18%.

|  |  | AG News-T | | AG News-D | | FPB | | Banking77 | | FMTOD | |
|---|---|---|---|---|---|---|---|---|---|---|---|
|  |  | Acc | F1 | Acc | F1 | Acc | F1 | Acc | F1 | Acc | F1 |
| Existing Model Set Difference | BM25 | 57.34 | 57.47 | 59.95 | 57.57 | 69.22 | 60.77 | 68.76 | 68.14 | 73.97 | 71.74 |
| | DPR | 56.50 | 54.45 | 56.98 | 54.76 | 66.67 | 58.06 | 68.70 | 68.06 | 71.67 | 68.43 |
| | TFIDF | 23.76 | 32.31 | 26.30 | 36.49 | 40.13 | 51.33 | 32.37 | 47.87 | 38.58 | 54.48 |
| | BERT | 71.39 | 59.49 | 75.63 | 65.18 | 79.05 | 70.69 | 77.84 | 68.15 | 89.99 | 85.42 |
| | RoBERTa | 70.35 | 58.11 | 77.15 | 67.22 | 77.90 | 69.46 | 75.10 | 64.49 | 86.41 | 80.44 |
| | Contriever | 75.24 | 64.62 | 79.93 | 71.06 | 77.07 | 68.86 | 79.53 | 70.52 | 87.75 | 82.52 |
| | SGPT | 67.85 | 54.87 | 67.85 | 54.86 | 76.71 | 67.34 | 69.35 | 56.84 | 89.95 | 85.38 |
| | SimCSE-BERT | 77.64 | 67.87 | 84.04 | 76.77 | 78.06 | 71.14 | 91.92 | 88.07 | 90.30 | 86.21 |
| | SimCSE-RoBERTa | 74.84 | 64.09 | 77.40 | 67.68 | 83.05 | 76.57 | 92.59 | 89.05 | 96.71 | 95.11 |
| | DiffCSE-BERT | 74.97 | 64.27 | 80.82 | 72.30 | 75.44 | 68.34 | 91.84 | 87.95 | 95.80 | 93.79 |
| | DiffCSE-RoBERTa | 71.64 | 59.87 | 78.40 | 68.96 | 80.86 | 73.68 | 93.43 | 90.27 | 96.59 | 94.92 |
| | MCSE-BERT | 74.72 | 63.96 | 83.24 | 75.68 | 78.46 | 71.23 | 88.43 | 83.03 | 96.66 | 95.03 |
| | MCSE-RoBERTa | 73.07 | 61.70 | 77.75 | 68.02 | 78.87 | 71.20 | 92.65 | 89.13 | 97.10 | 95.68 |
| SetCSE Difference | BERT | 87.39 | 81.54 | 92.92 | 89.55 | 84.91 | 78.14 | 97.22 | 95.87 | 99.42 | 99.14 |
| | RoBERTa | 89.35 | 84.36 | 85.12 | 78.50 | 86.86 | 80.85 | 94.31 | 91.73 | 99.70 | 99.55 |
| | Contriever | 85.93 | 79.56 | 92.95 | 89.61 | 83.50 | 76.27 | 95.12 | 92.78 | 98.94 | 98.42 |
| | SGPT | 67.99 | 55.04 | 68.36 | 55.53 | 77.73 | 68.63 | 70.59 | 58.44 | 93.39 | 90.21 |
| | SimCSE-BERT | 88.62 | 83.30 | **94.74** | **92.20** | 91.80 | 87.99 | 99.04 | 98.56 | **99.81** | **99.72** |
| | SimCSE-RoBERTa | **89.78** | **84.98** | **94.99** | **92.57** | **92.74** | **89.37** | 99.29 | 98.93 | 99.72 | 99.58 |
| | DiffCSE-BERT | 88.22 | 82.72 | 94.69 | 92.13 | 90.47 | 86.07 | 99.04 | 98.56 | **99.90** | **99.85** |
| | DiffCSE-RoBERTa | 89.01 | 83.87 | 94.31 | 91.58 | 91.94 | 88.22 | 99.14 | 98.72 | 98.95 | 98.43 |
| | MCSE-BERT | 88.33 | 82.89 | 93.84 | 90.89 | 91.96 | 88.21 | 98.94 | 98.41 | 99.76 | 99.64 |
| | MCSE-RoBERT | **89.86** | **85.09** | 94.36 | 91.65 | **93.19** | **90.00** | 99.14 | 98.72 | 99.63 | 99.45 |
| | Ave. Improvement | **19%** | **31%** | **18%** | **28%** | **15%** | **21%** | **12%** | **19%** | **6%** | **8%** |

Table 2: Evaluation results for SetCSE difference. As illustrated, the average improvements on accuracy and F1 are 14% and 21%, respectively.

As mentioned, the evaluation of SetCSE series of operations can be found in Appendiex C.4. The extensive evaluations indicate that SetCSE significantly enhances model discriminatory capabilities, and yields positive results in SetCSE intersection and SetCSE difference operations.

## 6 Application

As mentioned, SetCSE offers two significant advantages in information retrieval. One is its ability to effectively represent involved and sophisticated semantics, while the other is its capability to extract information associated with these semantics following complicated prompts. The former is achieved by expressing semantics with sets of sentences or phrases, and the latter is enabled by series of SetCSE intersection and difference operations. For instances, operation $A \cap B_1 \cap \cdots \cap B_N \setminus C_1 \setminus \cdots \setminus C_M$ essentially means "*to distinguish the difference between $B_1, \cdots, B_N, C_1, \cdots, C_M$, and to find sentences in $A$ that contains semantics in $B_i$'s while different from semantics $C_j$'s.*"

In this section, we showcase in detail these advantages through three important natural language processing tasks, namely, *complex and intricate semantic search*, *data annotation through active learning*, and *new topic discovery*. The datasets considered cover various domains, including financial analysis, legal service, and social media analysis. For more use cases and examples that are enabled by SetCSE, one can refer to Appendix D.

## 6.1 COMPLEX AND INTRICATE SEMANTIC SEARCH

In many real-world information retrieval tasks, there is a need to search for sentences with or without specific semantics that are hard to convey in single phrases or sentences. In these cases, existing querying methods based on single-sentence prompt are of limited use. By employing SetCSE, one can readily represent those semantics. Furthermore, SetCSE also supports expressing **convoluted** prompts via its operations and simple syntax. These advantages are illustrated through the following financial analysis example.

In recent years, there has been an increasing interest in leveraging a company's Environmental, Social, and Governance (ESG) stance to forecast its growth and sustainability (Utz, 2019; Hong et al., 2022). Brokerage firms and mutual fund companies have even begun offering financial products such as Exchange-Traded Funds (ETFs) that adhere to companies' ESG investment strategies (Kanuri, 2020; Rompotis, 2022) (more details on ESG are included in Appendix D.3). Notably, there is no definitive taxonomy for the term (CFA Institute, 2023), and lists of key topics are often used to illustrate these concepts (refer to Table 3a).

The intricate nature of ESG concepts makes it challenging to analyze ESG information through publicly available textual data, e.g., S&P 500 earnings calls (Qin & Yang, 2019). However, within the SetCSE framework, these can be readily represented by their example topics. Combined with several other semantics, one can effortlessly extract company earnings calls related to convoluted concepts such as "***using technology to solve Social issues, while neglecting its potential negative impact***," and "***investing in Environmental development projects***," through simple operations. Table 3 provides a detailed presentation of the corresponding SetCSE operations and results.

| | |
|---|---|
| Set $A$ - Environmental | {climate change, carbon emission reduction, water pollution, air pollution, renewable energy} |
| Set $B$ - Social | {diversity inclusion, community relations, customer satisfaction, fair wages, data security} |
| Set $C$ - Governance | {ethical practices, transparent accounting, business integrity, risk management, compliance} |
| Set $D$ - New Tech | {machine learning, artificial intelligence, robotics, generative model, neutral networks} |
| Set $E$ - Danger | {personal privacy breach, wrongful disclosure, pose threat, misinformation, unemployment} |
| Set $F$ - Invest | {strategic investement, growth investment, strategic plan, invest, investment} |

(a) Example topics for defining ESG (Investopedia, 2023) and example phrases for other semantics.

*Operation:* $X \cap B \cap D \setminus E$   `/*find sentence about "use tech to influence social issues positively"*/`

*Results:*

We're now using machine learning in most of our integrity work to keep our community safe.

But we know we also have a responsibility to deliver these fundamental technical and advances to fulfill the promise of bringing people closer together.

Our data and technology combined with specialized consulting experience help organization transition to a digital future while ensuring their workforce thrives.

And you'll see us integrating advances in machine learning so that customers can get better satisfaction

(b) Search for sentences related to "*using technology to solve Social issues, while neglecting its potential negative impact*," utilizing a serial of three SetCSE operations.

*Operation:* $X \cap A \cap F$   `/*find sentence about "invest in environmental development"*/`

*Results:*

We also continue to make progress on the $1.5 billion of undefined renewable prejects, which are included in our capital forecast.

To that end, our growth initiatives beyond the projects under construction have been focused on investments in natural gas and renewable projects with long term.

I would note that the $9.7 billion plan includes the natural gas storage as well as the UP generation investment that I just discussed.

(c) Search for sentences related to "*investing in Environmental development projects*," via simple SetCSE syntax.

Table 3: Demonstration of complex and intricate semantics search using SetCSE serial operations, through the example of analyzing S&P 500 company ESG stance leveraging earning calls transcripts.

## 6.2 DATA ANNOTATION AND ACTIVE LEARNING

Suppose building a classification model from scratch, and only an unlabeled dataset is present. Denote the unlabeled dataset as $X$, and each class as $i$, $i = 1, \cdots, N$. One quick solution is to use SetCSE as a filter to extract sentences that are semantically close to example set $S_i$ for each $i$, and then conduct a through human annotation, where the filtering is conducted using $X \cap S_i$. More interestingly, SetCSE supports *uncertainty labeling* in active learning framework (Settles, 2009; Gui et al., 2020), where the unlabeled items near a decision boundary between two classes $i$ and $j$ can be found using $X \cap S_i \cap S_j$.

We use Law Stack Exchange (LSE) (Li et al., 2022) dataset to validate the above data annotation strategy. Categories in this dataset include "copyright", "criminal law", "contract law", etc. Table 4a presents the sentences selected based on similarity with the example sets, whereas Table 4b shows the sentences that are on the decision boundaries between "copyright" and "criminal law". The latter indeed are difficult to categorize at first glance, hence labeling those items following the active learning framework would definitely increase efficiency in data annotation.

| *Operation:* $\boldsymbol{X \cap S_1}$ | /\*find sentences related to "copyright"\*/ |
|---|---|
| *Results:* | |
| Who owns a copyright on a scanned work? | |
| How does copyright on Recipes work? | |
| Is OCRed text automatically copyright? | |

| *Operation:* $\boldsymbol{X \cap S_2}$ | /\*find sentences related to "criminal law"\*/ |
|---|---|
| *Results:* | |
| Giving Someone Money Because of a Criminal Act? | |
| What are techniques used in law to robustly incentivize people to tell the truth? | |
| Canada - how long can a person be under investigation? | |

(a) Extract sentences close to "copyright" or "criminal law" categories for further human annotation.

| *Operation:* $\boldsymbol{X \cap S_1 \cap S_2}$ | /\*find sentences on decision boundary of "copyright" and "criminal law"\*/ |
|---|---|
| *Results:* | |
| Is there any country or state where the intellectual author of a homicide has twice or more the penalty than the physical author? | |
| Would police in the US have any alternative for handling a confiscated computer with a hidden partition? | |
| Is there a criminal database for my city Calgary | |
| Audio fingerprinting legal issues | |
| Is reading obscene written material online illegal in the UK? | |

(b) Following uncertainty labeling strategy in active learning framework, find sentences on the decision boundary between "copyright" and "criminal law" categories with the help of SetCSE serial operations.

Table 4: Demonstration of LSE dataset annotation and active learning utilizing SetCSE.

## 6.3 NEW TOPIC DISCOVERY

The task of new topic discovery (Blei & Lafferty, 2006; AlSumait et al., 2008; Chen et al., 2019) emerges when a dataset of interest is evolving over time. This can include tasks such as monitoring customer product reviews, collecting feedback for a currently airing TV series, or identifying trending public perception of specific stocks, among others. Suppose we have a unlabeled dataset $X$, and $N$ identified topics, the SetCSE operation for new topic extraction would be $X \setminus T_1 \setminus \cdots \setminus T_N$, where $T_i$ stands for the set of example sentences for topic $i$.

We use the Twitter Stance Evaluation datasets (Barbieri et al., 2020; Mohammad et al., 2018; Barbieri et al., 2018; Van Hee et al., 2018; Basile et al., 2019; Zampieri et al., 2019; Rosenthal et al., 2017; Mohammad et al., 2016) to illustrate the new topic discovery application. Specifically, we select "abortion", "etheism", and "feminist" as the existing topics, denoted as $T_1$, $T_2$, and $T_3$, and use $X \setminus T_1 \setminus T_2 \setminus T_3$ to find sentences with new topics. In the created evaluation dataset, the only other topic is "climate". As shown in Table 5, the top sentences extracted are indeed all related to this topic.

| *Operation:* $\boldsymbol{X} \setminus \boldsymbol{T_1} \setminus \boldsymbol{T_2} \setminus \boldsymbol{T_3}$ | `/*find sentences not related to "abortion", "etheism", or "feminist"*/` |
|---|---|
| *Results:* | |

@user Weather patterns evolving very differently over the last few years..

It's so cold and windy here in Sydney

On a scale of 1 to 10 the air quality in Whistler is a 35. #wildfires #BCwildfire #SemST

Look out for the hashtag #UKClimate2015 for news today on how the UK is doing in both reducing emissions and adapting to #SemST

Second heatwave hits NA NW popping up everywhere

Table 5: Demonstration of new topic discovery on Twitter leveraging SetCSE serial operations.

## 7   DISCUSSION

In this section, we provide quantitative justification of using sets to represent semantics, and comparison between SetCSE intersection and supervised learning. In addition, the performance of the embedding models post the context-specific inter-set contrastive learning are evaluated and presented in Appendix E.3.

Although the idea of expressing sophisticated semantics using sets instead of single sentences aligns with our intuition, its quantitative justification needs to be provided. We conduct experiments in Section 5, with $n_{\text{sample}}$ range from 1 to 30, where $n_{\text{sample}} = 1$ corresponds to querying by single sentences. The accuracy and F1 of those experiments can be found in Figures 4 and 8. As one can see, using sets ($n_{\text{sample}} > 1$) significantly improves querying performance. While $n_{\text{sample}} = 20$ would be sufficient to provide positive results in most of the cases.

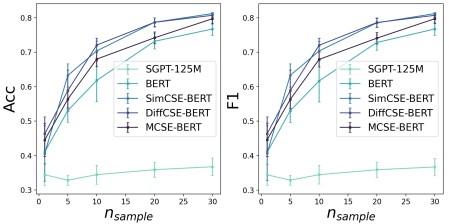
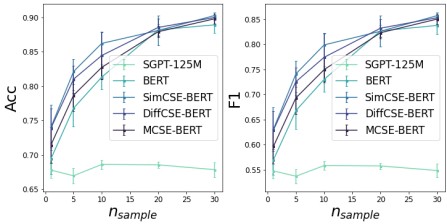



(a) SetCSE intersection performance.          (b) SetCSE difference performance.

Figure 4: SetCSE operation performances on AGT dataset for different values of $n_{\text{sample}}$.



We also compare the SetCSE intersection with supervised classification, where the latter regards sample sentences $Q_i$'s in Section 5.1 as training data, and predicts the semantics of $U$. Results of this evaluation can be found in Appendix E.2 and Table 13. As one can see, the performances are on par, while supervised learning results cannot be used in complex sentence querying tasks.

## 8   CONCLUSION AND FUTURE WORK

Taking inspiration from Set Theory, we introduce a novel querying framework named SetCSE, which employs sets to represent complex semantics and leverages its defined operations for structurally retrieving information. Within this framework, an inter-set contrastive learning objective is introduced. The efficacy of this learning objective in improving the discriminatory capability of the underlying sentence embedding models is demonstrated through extensive evaluations. The proposed SetCSE operations exhibit significant adaptability and utility in advancing information retrieval tasks, including complex semantic search, active learning, new topic discovery, and more.

Although we present comprehensive results in evaluation and application sections, there is still an unexplored avenue regarding testing SetCSE performance in various benchmark information retrieval tasks, applying the framework to larger embedding models for further performance improvement, and potentially incorporating LoRA into the framework (Hu et al., 2021). Additionally, we aim to create a SetCSE application interface that enables quick sentence extraction through its straightforward syntax.

ACKNOWLEDGEMENT

We express gratitude to Di Xu, Cong Liu, Yu-Ching Shih, and Hsi-Wei Hsieh for their enlightening discussions. Additionally, we extend our thanks to the anonymous area chair and reviewers for their constructive comments and suggestions.

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

# APPENDIX

# Table of Contents

# A    RELATED WORK

## A.1    SET THEORY IN FORMAL SEMANTICS

At its core, Formal Semantics aims to create precise, rule-based systems that capture the meaning of language constructs, from words and phrases to complex sentences and discourse (Chierchia & McConnell-Ginet, 1990; Cann, 1993; Partee, 2005; Portner & Partee, 2008; Partee et al., 2012). Set theory plays a pivotal role in achieving this goal. In particular, we highlight the following contributions of Set Theory to Formal Semantics mentioned in Portner & Partee (2008):

**Semantic Representation.** Set theory is used to represent the meanings of words and phrases. Individual elements of sets can represent various semantic entities, such as objects, actions, or properties. For example, the set dog might represent the concept of a dog, while the set run represents the action of running.

**Compositionality.** One of the fundamental principles in Formal Semantics is compositionality, which states that the meaning of a complex expression is determined by the meanings of its parts and how they are combined.

**Predicate Logic.** Set theory often integrates with predicate logic to represent relationships and quantification in natural language. Predicate logic allows for the representation of propositions, and set theory complements this by representing the sets of entities that satisfy these propositions.

## A.2    SET OPERATIONS FOR WORD INTERPRETATION AND EMBEDDING IMPROVEMENT

Section 2 highlights several works that have employed set operations on word embeddings to interpret relationships between words, leading to quantitative and qualitative improvements in word embedding qualities (Zhelezniak et al., 2019; Bhat et al., 2020; Dasgupta et al., 2021).

The novelty of the SetCSE framework, which utilizes embeddings and set-theoretic operations, is specifically addressed in relation to the aforementioned works below:

1. SetCSE employs **sentence embeddings** for semantic representation and information retrieval, diverging from the prior focus of the mentioned works on using and improving **word embeddings**.

2. SetCSE utilizes sets of sentences and its learning mechanism to recognize and represent complex and intricate semantics for information querying. This approach differs from previous works, which did not consider the collective use of words to represent complex semantics.

3. SetCSE integrates set-theoretic operations for expressing complex queries in practical sentence retrieval tasks, distinguishing it from previous works that used set operations to uncover word relationships.

# B    SETCSE OPERATIONS

## B.1    PROPERTIES OF SETCSE OPERATIONS

Note that, as per Definition 2, the output of SetCSE serial operations, $A \cap B_1 \cap \cdots \cap B_N \backslash D \backslash \cdots \backslash D_M$, forms an ordered set of elements in $A$. Hence, these operations aren't strictly equivalent to the operations defined in Set Theory (Cantor, 1874; Johnson-Laird, 2004), lacking certain properties of the latter, such as the commutative law. Despite this asymmetry, the definitions within the SetCSE framework offer several advantages:

- It is intuitive to borrow the concepts of intersection and difference operations to describe the "selection" and "deselection" of sentences with certain semantics.

- Serving as a querying framework, SetCSE is designed to retrieve information from a set of sentences following certain queries. And the proposed SetCSE operation syntax aligns well with its purpose. For instance, the SetCSE serial operations $A \cap B \backslash C$ means "*finding sentences in the set $A$ that contains the semantics $B$ but not $C$.*"

# C EVALUATION

## C.1 HYPERPARAMETER OPTIMIZATION

The effect of temperature parameter $\tau$ and training epoch can be found in Tables 6 and 7, respectively. The effect of using different $n_{sample}$ to represent semantics is discussed in Section 7. In particular, when optimizing for $\tau$ and training epoch, we consider SimCSE-BERT model, AGT dataset and SetCSE intersection operation.

| $\tau$ | 0.001 | 0.01 | 0.05 | 0.1 | 1 |
|---|---|---|---|---|---|
| Acc | 77.14 | 77.31 | 78.29 | 77.42 | 77.54 |
| F1 | 77.11 | 77.29 | 78.27 | 77.40 | 77.52 |

Table 6: Effects of different temperature $\tau$ for SetCSE intersection on AGT dataset.

| Epoch | 20 | 30 | 40 | 50 | 60 | 70 | 80 | 90 |
|---|---|---|---|---|---|---|---|---|
| Acc | 71.40 | 74.63 | 75.76 | 75.82 | 78.27 | 79.25 | 79.74 | 80.47 |
| F1 | 71.42 | 74.54 | 75.66 | 75.70 | 78.22 | 79.16 | 79.72 | 80.41 |

Table 7: Effects of different training epoch for SetCSE intersection on AGT dataset.

## C.2 THE T-SNE PLOTS OF SENTENCE EMBEDDINGS

As previously mentioned, to illustrate the SetCSE framework performance in a more intuitive manner, we include the t-SNE (Van der Maaten & Hinton, 2008) plots of the sentence embeddings regarding all dataset considered. As one can see, the improvements on AGT, AGD and FPB datasets are significant, while the improvements on FMTOD is smaller, since for the latter, the underlying semantics are distinctive already.

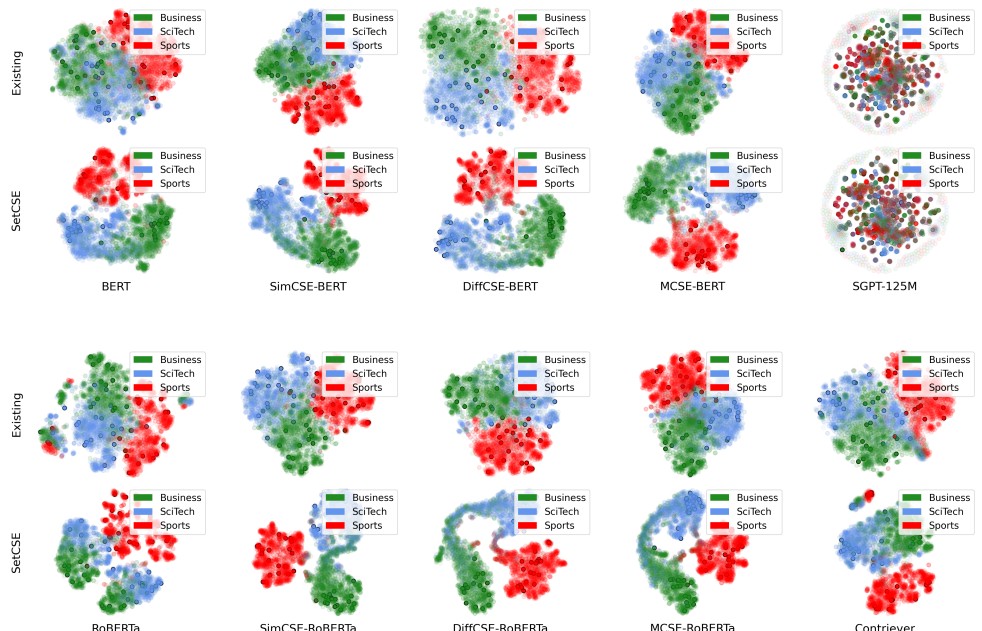

Figure 5: The t-SNE plots of sentence embeddings induced by existing language models and the SetCSE fine-tuned ones for AGD dataset. As illustrated, the model awareness of different semantics are significantly improved.

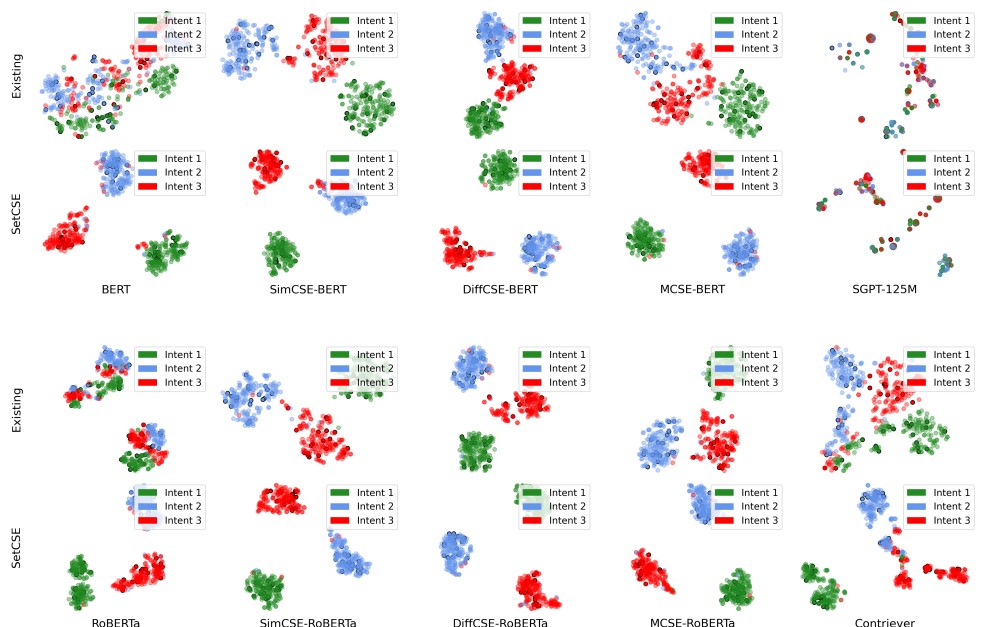

Figure 6: The t-SNE plots of sentence embeddings induced by existing language models and the SetCSE fine-tuned ones for Banking77 dataset, where "Intent 1", "Intent 2" and "Intent 3" are "card payment fee charged", "direct debit payment not recognised" and "balance not updated after cheque or cash deposit", respectively. As illustrated, the improvements of model awareness of different semantics can be observed.

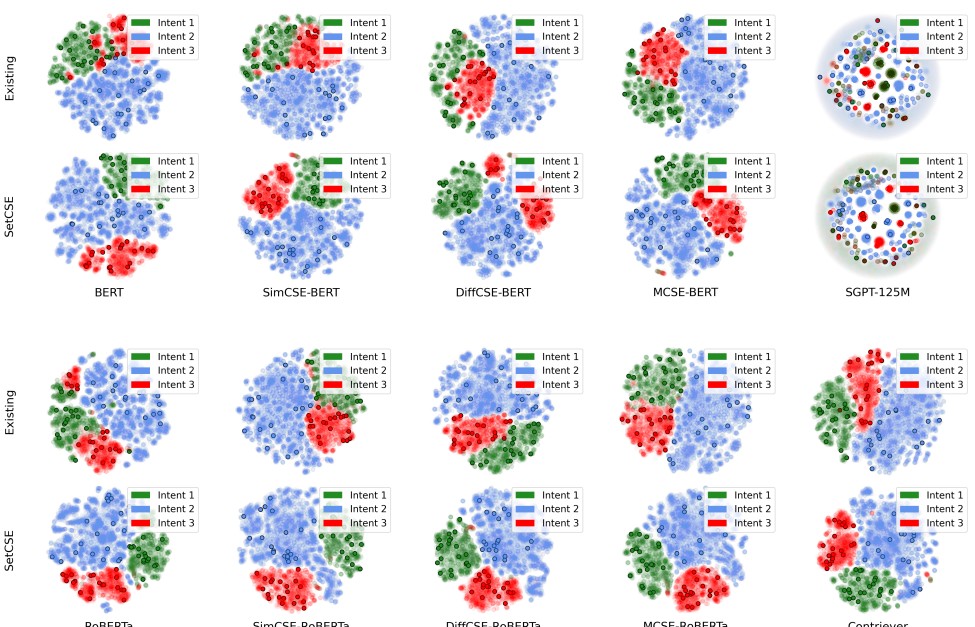

Figure 7: The t-SNE plots of sentence embeddings induced by existing language models and the SetCSE fine-tuned ones for FMTOD dataset, where "Intent 1", "Intent 2" and "Intent 3" are "find weather", "set alarm" and "set reminder", respectively. As illustrated, the improvements of model awareness of different semantics are not as prominent as the ones with other datasets, which aligns with Table 1 results.

## C.3 Discussion on SGPT Performance

In Section 5, our evaluations demonstrate that the decoder-only SGPT-125M performs less effectively compared to encoder-based models of similar sizes, both before and after inter-set contrastive learning stages. This observation aligns with findings from other studies that compare embeddings produced by BERT-based models and GPT in benchmark word embedding tasks (Ethayarajh, 2019; Liu et al., 2020; Cai et al., 2020).

Since our evaluation indicates that SGPT benefits less from inter-set fine-tuning, future studies may consider other contrastive learning methods (Jian et al., 2022; Jain et al., 2023) to enhance the context awareness and discriminatory capabilities of decoder-based models.

## C.4 Evaluation for SetCSE Serial Operations

In this section, we evaluate the performance of SetCSE serial operations. Specifically, we consider the following three serial operations:

- Series of two SetCSE intersection operations.
- Series of two SetCSE difference operations.
- Series of SetCSE intersection and difference operations.

We utilize multi-label datasets to conduct the SetCSE serial operations experiment. To encompass diverse contexts, we consider the following multi-label datasets and their semantics:

- GitHub Issue (GitHub) (Ismael, 2022) — "*help wanted*" (H), "*docs*" (D).
- English Quotes (Quotes) (Eltaief, 2022) — "*inspirational*" (I), "*love*" (L), "*life*" (F).
- Reuters-21578 (Reuters) (Lewis, 1997) — "*ship*" (S), "*grain*" (G), "*crude*" (C).

### C.4.1 Evaluation of SetCSE Intersection Series

Suppose a multi-label dataset $S$ has $N$ semantics, where $S_i$ denotes the set of sentences with the $i$-th semantic, and each sentence in $S$ contains several semantics in the set of $\{1, \ldots, N\}$. For evaluating two serial SetCSE intersections, the experiment is set up as follows:

1. For $S_i$, randomly select $n_{\text{sample}}$ of sentences, denoted as $Q_i$, and concatenate remaining sentences in all $S_i$, denoted as $U$. Regard $Q_i$'s as example sets and $U$ as the evaluation set.
2. Select two sample sets $Q_i$ and $Q_j$, $i, j \in \{1, \ldots, N\}$, and conduct $U \cap Q_i \cap Qj$ following Algorithm 1. Select the top $|U_{i,j}|$ from the results of serial operations, where $U_{i,j} \subseteq U$ denotes the set of sentences containing semantics $i$ and $j$. Predict the selected sentences containing semantics $i$ and $j$, and compare against ground truth to compute accuracy and F1.
3. As a control group, repeat Step 2 while omitting the model fine-tuning in Algorithm 1.
4. To compare with the performance of single SetCSE operation, conduct experiment in Subsection 5.2 for semantics $i$ and $j$ on $U$.

The parameters utilized in the experiments within this section remain consistent with those employed in Section 5.1. Detailed results pertaining to the above experiment can be found in Table 8. For instance, within the "GitHub-HD" column, the results are presented utilizing the GitHub dataset, with semantics $i$ and $j$ designated as "*help wanted*" and "*docs*", respectively. Notably, the SetCSE framework showcases a 26% improvement in the performance of serial intersections. Additionally, it is observed that the accuracy and F1 scores of two consecutive SetCSE intersections closely approximate the product of the accuracy and F1 scores from two separate SetCSE intersections, respectively.

### C.4.2 Evaluation of SetCSE Difference Series

Suppose a multi-label dataset $S$ has $N$ semantics, where $S_i$ denotes the set of sentences with the $i$-th semantic, and each sentence in $S$ contains several semantics in the set of $\{1, \ldots, N\}$. For evaluating two serial SetCSE intersections, the experiment is set up as follows:

1. For $S_i$, randomly select $n_{\text{sample}}$ of sentences, denoted as $Q_i$, and concatenate remaining sentences in all $S_i$, denoted as $U$. Regard $Q_i$'s as example sets and $U$ as the evaluation set.

| | | GitHub-HD | | Quotes-FL | | Quotes-FI | | Reuters-SG | | Reuters-SC | |
|---|---|---|---|---|---|---|---|---|---|---|---|
| | | Acc | F1 | Acc | F1 | Acc | F1 | Acc | F1 | Acc | F1 |
| Existing Model Single Intersection | BERT | 71.93 | 75.53 | 83.02 | 86.00 | 91.12 | 92.00 | 86.94 | 89.60 | 91.77 | 92.57 |
| | RoBERTa | 71.68 | 75.30 | 83.85 | 86.90 | 90.71 | 91.68 | 87.50 | 90.01 | 90.26 | 91.37 |
| | Contriever | 75.81 | 78.67 | 84.85 | 87.86 | 92.15 | 92.83 | 79.30 | 81.85 | 92.81 | 93.42 |
| | SimCSE-BERT | 74.20 | 74.62 | 89.80 | 91.87 | 90.63 | 91.60 | 92.27 | 93.40 | 90.36 | 91.49 |
| | DiffCSE-BERT | 74.21 | 77.47 | 88.52 | 93.76 | 90.53 | 91.23 | 92.90 | 93.08 | 90.75 | 91.50 |
| | MCSE-BERT | 74.56 | 77.75 | 90.08 | 91.93 | 90.51 | 91.49 | 91.32 | 92.27 | 91.07 | 92.02 |
| SetCSE Single Intersection | BERT | 87.27 | 86.44 | 92.70 | 93.84 | 93.16 | 93.70 | 96.38 | 96.68 | 95.92 | 96.12 |
| | RoBERTa | 87.77 | 89.16 | 90.93 | 92.54 | 94.08 | 94.54 | 95.18 | 95.68 | 95.88 | 96.12 |
| | Contriever | 92.34 | 93.38 | 92.76 | 93.85 | 94.92 | 95.22 | 95.47 | 95.96 | 93.42 | 94.01 |
| | SimCSE-BERT | 93.77 | 94.48 | 92.58 | 93.74 | 92.09 | 92.82 | 96.72 | 96.95 | 95.89 | 96.11 |
| | DiffCSE-BERT | 92.11 | 94.10 | 92.57 | 93.01 | 91.23 | 92.67 | 95.84 | 96.21 | 94.66 | 94.97 |
| | MCSE-BERT | 91.67 | 92.91 | 92.15 | 93.42 | 92.46 | 93.13 | 96.93 | 97.19 | 95.78 | 96.10 |
| Existing Model Serial Intersections | BERT | 33.54 | 36.73 | 54.37 | 57.00 | 78.21 | 79.94 | 57.85 | 59.14 | 79.75 | 81.78 |
| | RoBERTa | 38.10 | 41.45 | 49.17 | 52.42 | 79.72 | 81.37 | 57.73 | 62.60 | 78.09 | 80.44 |
| | Contriever | 47.78 | 53.37 | 51.35 | 54.48 | 83.05 | 84.47 | 57.78 | 59.36 | 82.77 | 84.23 |
| | SimCSE-BERT | 48.33 | 51.02 | 65.47 | 68.57 | 78.17 | 80.52 | 72.82 | 74.14 | 79.05 | 81.19 |
| | DiffCSE-BERT | 49.71 | 51.66 | 65.79 | 68.33 | 79.33 | 82.80 | 69.33 | 73.43 | 80.02 | 82.43 |
| | MCSE-BERT | 48.67 | 51.01 | 66.20 | 67.84 | 79.69 | 81.70 | 69.54 | 73.94 | 81.15 | 82.92 |
| SetCSE Serial Intersections | BERT | 58.33 | 62.28 | 69.15 | 73.88 | 83.90 | 85.23 | 85.53 | 86.74 | 92.09 | 92.41 |
| | RoBERTa | 53.66 | 58.67 | 64.32 | 67.32 | 85.95 | 87.07 | 84.66 | 86.00 | 92.38 | 92.62 |
| | Contriever | 71.74 | 75.52 | 71.32 | 74.24 | **89.55** | **90.11** | 82.22 | 84.07 | 84.25 | 85.66 |
| | SimCSE-BERT | **76.93** | **79.53** | **72.22** | **74.54** | 88.22 | 89.68 | **90.12** | **90.63** | **92.55** | **92.87** |
| | DiffCSE-BERT | 75.26 | 76.16 | 70.31 | 73.62 | 84.08 | 85.75 | 88.85 | 90.77 | 90.59 | 91.97 |
| | MCSE-BERT | 76.83 | 77.43 | 71.48 | 72.50 | 82.99 | 84.49 | 88.39 | 89.30 | 91.05 | 91.77 |
| | Ave. Improvement | **55%** | **51%** | 22% | 22% | 8% | 7% | **38%** | **34%** | 13% | 11% |

Table 8: Evaluation results for series of two SetCSE intersection operations. As illustrated, the average improvements on accuracy and F1 are 27% and 25%, respectively.

2. Select two sample sets $Q_i$ and $Q_j$, $i, j \in \{1, \dots, N\}$, and conduct $U \setminus Q_i \setminus Qj$ following Algorithm 1. Select the top $|U_{\bar{i},\bar{j}}|$ from the results of serial operations, where $U_{\bar{i},\bar{j}} \subseteq U$ denotes the set of sentences that do not contain either semantics $i$ or $j$. The selected sentences are predicted as not containing either semantics $i$ or $j$, and accuracy and F1 are calculated against the ground truth.

3. As a control group, repeat Step 2 while omitting the model fine-tuning in Algorithm 1.

4. To compare with the performance of single SetCSE operation, conduct experiment in Subsection 5.1 for semantics $i$ and $j$ on $U$.

The detailed experiment results can be found in Table 9. As one can see, the SetCSE framework improves performance of serial difference operations by 37%. Similarly to Section C.4.1, it is observed that the accuracy and F1 scores of two consecutive SetCSE difference operations closely approximate the product of the accuracy and F1 scores from two separate SetCSE difference operations, respectively.

### C.4.3 Evaluation of SetCSE Intersection and Difference Series

Suppose a multi-label dataset $S$ has $N$ semantics, where $S_i$ denotes the set of sentences with the $i$-th semantic, and each sentence in $S$ contains several semantics in the set of $\{1, \dots, N\}$. For evaluating two serial SetCSE intersections, the experiment is set up as follows:

1. For $S_i$, randomly select $n_{\text{sample}}$ of sentences, denoted as $Q_i$, and concatenate remaining sentences in all $S_i$, denoted as $U$. Regard $Q_i$'s as example sets and $U$ as the evaluation set.

2. Select two sample sets $Q_i$ and $Q_j$, $i, j \in \{1, \dots, N\}$, and conduct $U \cap Q_i \setminus Qj$ following Algorithm 1. Select the top $|U_{i,\bar{j}}|$ from the results of serial operations, where $U_{i,\bar{j}} \subseteq U$ denotes the set of sentences containing semantics $i$ but not $j$. The selected sentences are predicted as the ones containing semantics $i$ but not $j$, and the accuracy and F1 are calculated against the ground truth.

3. As a control group, repeat Step 2 while omitting the model fine-tuning in Algorithm 1.

4. To compare with the performance of single SetCSE operation, conduct experiment in Subsection 5.1 and 5.2 for semantics $i$ and $j$ on $U$, respectively.

The detailed experiment results can be found in Table 9. As one can see, the SetCSE framework improves performance of serial difference operations by 35%. Similarly to Sections C.4.1 and

| | | GitHub-HD | | Quotes-FL | | Quotes-FI | | Reuters-SG | | Reuters-SC | |
|---|---|---|---|---|---|---|---|---|---|---|---|
| | | Acc | F1 | Acc | F1 | Acc | F1 | Acc | F1 | Acc | F1 |
| Existing Model Single Difference | BERT | 71.38 | 72.43 | 65.77 | 71.19 | 65.30 | 70.86 | 80.36 | 82.35 | 77.88 | 80.49 |
| | RoBERTa | 70.90 | 71.95 | 66.34 | 71.59 | 64.49 | 70.60 | 81.74 | 83.45 | 79.23 | 81.54 |
| | Contriever | 71.83 | 72.92 | 66.89 | 71.97 | 66.00 | 71.34 | 73.93 | 77.52 | 73.56 | 77.25 |
| | SimCSE-BERT | 71.45 | 72.73 | 72.78 | 76.32 | 68.61 | 71.80 | 83.79 | 85.10 | 81.27 | 83.07 |
| | DiffCSE-BERT | 71.49 | 72.08 | 73.78 | 75.71 | 68.84 | 71.23 | 85.94 | 86.09 | 83.06 | 84.51 |
| | MCSE-BERT | 71.34 | 72.56 | 72.22 | 75.91 | 68.37 | 70.91 | 87.40 | 88.21 | 84.74 | 86.01 |
| SetCSE Single Difference | BERT | 81.88 | 82.64 | 75.69 | 78.65 | 71.15 | 73.64 | 96.96 | 97.01 | 96.15 | 96.23 |
| | RoBERTa | 83.16 | 84.28 | 71.87 | 75.91 | 69.66 | 72.61 | 96.03 | 96.13 | 95.25 | 95.38 |
| | Contriever | 85.27 | 86.11 | 79.63 | 81.85 | 73.91 | 75.66 | 95.58 | 95.69 | 94.09 | 94.27 |
| | SimCSE-BERT | 87.39 | 88.25 | 80.02 | 82.13 | 72.03 | 75.77 | 97.13 | 97.18 | 96.44 | 96.51 |
| | DiffCSE-BERT | 86.23 | 87.19 | 81.51 | 81.87 | 73.30 | 75.76 | 98.20 | 98.37 | 96.50 | 96.78 |
| | MCSE-BERT | 86.06 | 87.13 | 80.06 | 82.20 | 73.99 | 75.72 | 97.37 | 97.41 | 96.41 | 96.48 |
| Existing Model Serial Difference | BERT | 41.20 | 42.36 | 42.31 | 45.69 | 48.67 | 51.32 | 57.58 | 58.96 | 58.74 | 61.24 |
| | RoBERTa | 42.06 | 45.07 | 42.93 | 46.27 | 49.02 | 51.61 | 61.66 | 62.61 | 60.10 | 62.91 |
| | Contriever | 42.61 | 45.07 | 38.60 | 42.06 | 51.70 | 53.77 | 56.28 | 58.01 | 56.39 | 57.36 |
| | SimCSE-BERT | 42.16 | 44.22 | 50.04 | 52.42 | 50.34 | 52.67 | 64.73 | 66.17 | 66.49 | 68.11 |
| | DiffCSE-BERT | 42.11 | 43.91 | 50.37 | 52.20 | 47.58 | 50.09 | 66.17 | 66.83 | 67.30 | 71.26 |
| | MCSE-BERT | 42.41 | 44.71 | 51.27 | 53.41 | 46.71 | 49.66 | 66.03 | 68.50 | 70.32 | 73.01 |
| SetCSE Serial Difference | BERT | 51.13 | 52.23 | 48.48 | 50.93 | 63.47 | 65.26 | 91.84 | 91.99 | 91.10 | 91.18 |
| | RoBERTa | 61.15 | 62.26 | 45.60 | 48.26 | 56.31 | 56.55 | 90.34 | 90.58 | 91.71 | 91.85 |
| | Contriever | 65.04 | 69.46 | 53.22 | 54.85 | 69.58 | 71.23 | 89.10 | 89.43 | 90.24 | 90.48 |
| | SimCSE-BERT | 65.36 | 70.01 | **55.54** | **56.67** | 67.03 | 69.91 | 93.05 | 93.14 | 92.67 | 92.77 |
| | DiffCSE-BERT | 66.90 | 67.29 | 53.42 | 56.01 | 69.11 | 71.45 | 92.64 | 93.19 | 91.45 | 92.22 |
| | MCSE-BERT | **67.17** | **73.20** | 53.54 | 55.15 | **69.67** | **71.80** | **93.10** | **93.19** | **92.91** | **93.01** |
| | Ave. Improvement | 47% | 45% | 15% | 12% | 32% | 29% | **50%** | **47%** | **49%** | **44%** |

Table 9: Evaluation results for series of two SetCSE difference operations. As illustrated, the average improvements on accuracy and F1 are 38% and 35%, respectively.

C.4.2, we observe that the accuracy and F1 scores of two consecutive SetCSE intersections closely approximate the product of the accuracy and F1 scores from two separate SetCSE intersections, respectively.

| | | GitHub-HD | | Quotes-FL | | Quotes-FI | | Reuters-SG | | Reuters-SC | |
|---|---|---|---|---|---|---|---|---|---|---|---|
| | | Acc | F1 | Acc | F1 | Acc | F1 | Acc | F1 | Acc | F1 |
| Existing Model Single Operation | BERT | 85.22 | 87.80 | 77.74 | 79.40 | 77.52 | 80.02 | 80.17 | 81.27 | 78.41 | 80.62 |
| | RoBERTa | 83.63 | 86.70 | 78.22 | 79.75 | 76.58 | 79.16 | 80.15 | 81.26 | 77.66 | 80.10 |
| | Contriever | 88.06 | 89.84 | 78.47 | 79.93 | 79.00 | 81.27 | 74.77 | 77.25 | 80.55 | 82.13 |
| | SimCSE-BERT | 86.82 | 88.94 | 92.50 | 93.14 | 86.54 | 88.87 | 94.45 | 94.81 | 87.07 | 89.59 |
| | DiffCSE-BERT | 86.18 | 88.34 | 92.33 | 93.37 | 86.04 | 88.21 | 94.61 | 95.42 | 88.43 | 90.76 |
| | MCSE-BERT | 86.63 | 88.80 | 91.83 | 92.59 | 86.64 | 89.18 | 93.77 | 94.22 | 89.08 | 91.08 |
| SetCSE Single Operation | BERT | 91.27 | 92.31 | 94.25 | 94.68 | 89.47 | 91.50 | 97.80 | 97.86 | 97.62 | 97.74 |
| | RoBERTa | 88.99 | 90.53 | 93.56 | 94.13 | 92.04 | 93.43 | 97.47 | 97.58 | 97.31 | 97.48 |
| | Contriever | 90.35 | 91.60 | 93.85 | 94.32 | 92.30 | 93.70 | 97.84 | 97.91 | 92.78 | 93.97 |
| | SimCSE-BERT | 94.13 | 94.59 | 95.45 | 95.70 | 92.16 | 93.61 | 98.51 | 98.54 | 98.37 | 98.43 |
| | DiffCSE-BERT | 93.55 | 96.32 | 94.42 | 95.39 | 93.17 | 93.77 | 97.56 | 98.18 | 97.30 | 97.66 |
| | MCSE-BERT | 93.38 | 93.99 | 94.70 | 95.04 | 91.88 | 93.40 | 97.49 | 97.58 | 97.21 | 97.40 |
| Existing Model Serial Operations | BERT | 59.40 | 66.64 | 62.85 | 66.36 | 59.60 | 61.76 | 52.36 | 54.26 | 63.62 | 65.40 |
| | RoBERTa | 56.84 | 62.84 | 63.46 | 66.82 | 58.89 | 60.80 | 58.18 | 58.68 | 59.94 | 61.36 |
| | Contriever | 59.43 | 63.81 | 63.91 | 67.17 | 57.57 | 59.15 | 58.96 | 61.53 | 62.90 | 69.13 |
| | SimCSE-BERT | 63.79 | 69.74 | 80.76 | 82.59 | 64.45 | 66.65 | 66.59 | 68.64 | 68.41 | 70.32 |
| | DiffCSE-BERT | 63.96 | 65.64 | 82.89 | 83.08 | 64.48 | 65.93 | 67.70 | 68.82 | 66.74 | 68.19 |
| | MCSE-BERT | 64.24 | 70.06 | 81.99 | 83.59 | 63.26 | 65.98 | 68.46 | 70.71 | 67.97 | 69.20 |
| SetCSE Serial Operations | BERT | 76.43 | 79.23 | 86.11 | 87.16 | 82.33 | 83.44 | 91.58 | 91.73 | 90.05 | 90.55 |
| | RoBERTa | 70.38 | 74.52 | 84.49 | 85.86 | 86.43 | 88.02 | 90.80 | 91.08 | 88.50 | 89.25 |
| | Contriever | 74.00 | 77.35 | 85.18 | 86.32 | **87.73** | **88.46** | 91.67 | 91.85 | 90.47 | 92.31 |
| | SimCSE-BERT | **84.17** | **85.42** | **89.02** | **89.64** | 87.27 | 88.12 | **93.36** | **93.43** | **93.02** | **93.32** |
| | DiffCSE-BERT | 82.17 | 84.82 | 87.46 | 88.92 | 86.53 | 88.20 | 91.65 | 92.68 | 90.28 | 90.28 |
| | MCSE-BERT | 82.07 | 83.73 | 87.21 | 88.02 | 86.01 | 88.24 | 90.85 | 91.06 | 88.26 | 89.04 |
| | Ave. Improvement | 27% | 22% | 24% | 21% | **41%** | **39%** | **52%** | **49%** | 41% | 37% |

Table 10: Evaluation results for series of SetCSE intersection and difference operations. As illustrated, the average improvements on accuracy and F1 are 37% and 33%, respectively.

The experiments presented in this section indicate that, on average, SetCSE improves the performance of serial operations by 33%. Combining with the evaluation results of Section 5, we conclude that SetCSE significantly enhances model discriminatory capabilities, and yields positive results in SetCSE intersection, difference, and series of operations.

# D   APPLICATION

## D.1   ERROR ANALYSIS FOR APPLICATION CASE STUDIES

To further illustrate the performance and stability of SetCSE serial operations in practical applications, we conduct an error analysis on the showcased examples in Section 6.1. Specifically, Tables 11a and 11b present the less preferable query results compared to Tables 3b and 3c, respectively. It's important to note that the presented results are not ranked among the top sentences in the corresponding SetCSE querying outputs. Instead, they are listed between the 30th to 50th positions within the sentences of the S&P500 earnings call dataset (Qin & Yang, 2019).

---

*Operation:* $X \cap B \cap D \setminus E$   `/*find sentence about "use tech to influence social issues positively"*/`

*Error Analysis:*

It is fairly incremental in terms of adding things like customer support, field application engineering, software support, given that we're familiarizing people with our architecture.

We had good bio-security to begin with, but we amped it up to an all-time high level of discipline and scrutiny, frankly, and it hasn't stopped.

We are in a unique position to combine the state-of-the-art online experience with the exceptional customer service our associates are known for.

Finally, there is one commonality our customers have, it's that they live in a hybrid IT world.

---

(a) Error analysis for complex semantic search with the query "*using technology to solve Social issues, while neglecting its potential negative impact.*"

---

*Operation:* $X \cap A \cap F$   `/*find sentence about "invest in environmental development"*/`

*Error Analysis:*

But in terms of percentage growth, most of it's going to come from gas we would expect.

Although energy storage has significant potential for growth, at this point, we have not assumed any material contributions in our outlook.

We expect the pace of reduction in loan balances to slow up as energy prices have stabilized and the rig count has increased.

---

(b) Error analysis for complex semantic search with the query "*investing in environmental development projects.*"

Table 11: Error analysis of complex and intricate semantics search using SetCSE serial operations, through the example of analyzing S&P 500 company ESG stance leveraging earning calls transcripts.

Based on the findings in Table 11, we identify the following issues that may lead to less preferred results:

- Certain words in the query results closely match sample sentences in terms of represented semantics, leading to their respective rankings. However, the entire sentences show less relevance to those specific semantics. For instance, word "*energy*" may align with the sample phrase "*renewable energy*" yet the entire sentence might be less associated with "*Environmental issues*".

- Some sentences are chosen due to their high relevance to a single presented semantic, while an intersection of multiple semantics is anticipated. For example, the sentence "*We had good **bio-security** to begin with, but we amped it up to an all-time high level of **discipline and scrutiny**, frankly, and it hasn't stopped.*" aligns closely with "*Social issue*" but displays less relevance to "*new technology*".

It's essential to note that the mentioned examples are not ranked among the top sentences in the query outputs, thereby naturally leading to errors. We aim to improve the assessment of the closeness between a sentence and sets of semantics to mitigate some of these aforementioned errors.

## D.2   ADDITIONAL USE CASES AND EXAMPLES USING SETCSE

**Data Pre-labeling.** Besides Section 6.2, one additional example of leveraging SetCSE to preprocess unlabeled data is included in Table 12, where Banking77 dataset is used.

| *Operation:* $X \cap I_1$ | /*find sentences related to "card payment fee charged"*/ |
|---|---|

*Results:*

Why was I charged an extra fee when using the card?

How come I was charged an extra fee when paying with the card?

I paid with my card and got charged an extra fee, what's up with that

Is it normal to be charged an extra fee when paying with my card?

| *Operation:* $X \cap I_2$ | /*find sentences related to "balance not updated after cheque or cash deposit"*/ |
|---|---|

*Results:*

How long should a cheque deposit take to show? My account hasn't updated and I want to make sure everything is okay.

My balance is not right. It has not been updated for the cash or cheque deposit.

I made a cash deposit a few days ago and it's still not reflected in my account. Do you know what might have happened?

I attempted to deposit a cheque yesterday but the balance isn't showing today. Is it still pending?

Table 12: Demonstration of Banking77 dataset pre-labeling utilizing SetCSE. Specifically, $I_1$ and $I_2$ denote the sample sets for categories "card payment fee charged" and "balance not updated after cheque or cash deposit".

**Word Sense Disambiguation.** Word Sense Disambiguation (WSD) is a fundamental task in natural language processing and computational linguistics. It refers to the process of determining the correct sense or meaning of a word when that word has multiple possible meanings or senses in a particular context (Agirre & Edmonds, 2007; Navigli, 2009; Bevilacqua et al., 2021). Using single prompt that contains these polysemies for information retrieval often yields unsatisfactory results, while one can use SetCSE to represent the exact meaning through multiple phrases or sentences and conduct information extraction.

## D.3 INTRODUCTION TO ESG

ESG stands for Environmental, Social, and Governance, and it is a framework used to evaluate and measure the sustainability and ethical practices of a company or organization. ESG criteria are used by investors, analysts, and stakeholders to assess how a company manages its impact on the environment, its relationships with society, and the quality of its corporate governance (Ide & Véronis, 1998; Stevenson & Wilks, 2003; McCarthy, 2009; Friede et al., 2015; Reiser & Tucker, 2019; Li et al., 2021; Khan, 2022; Arvidsson & Dumay, 2022). From a corporate standpoint, enhancing the ESG footprint is equally advantageous as investing directly in productivity and automation (Abiri et al., 2017; Alavian et al., 2019; Liu et al., 2019a; Kazekami, 2020; Alavian et al., 2020; Liu, 2021; Eun et al., 2022; Alavian et al., 2022; Chui et al., 2023), as it aids in talent attraction and fosters long-term sustainability (Henisz et al., 2019; Woo & Tan, 2022).

According to Investopedia (2023) and CFA Institute (2023), the term ESG includes but not limited to the following topics.

**Environmental.** This aspect focuses on a company's environmental impact and its efforts to address sustainability challenges, its key topics include: *Climate policies*, *Energy use*, *Waste*, *Pollution*, *Natural resource conservation*, *Treatment of animals*.

**Social.** The social component of ESG centers on how a company manages its relationships with people and communities, its key factors include: *Customer satisfaction*, *Data protection and privacy*, *Gender and diversity*, *Employee engagement*, *Community relations*, *Human rights*, *Labor standards*.

**Governance.** This aspect focuses on the internal governance and management practices of a company, its key factors include: *Board composition*, *Audit committee structure*, *Bribery and corruption*, *Executive compensation*, *Lobbying*, *Political contributions*, *Whistle-blower schemes*.

# E DISCUSSION

## E.1 JUSTIFICATION OF LEVERAGING SETS TO REPRESENT SEMANTICS

As previously mentioned, we conduct experiments in Section 5, with $n_{\text{sample}}$ range from 1 to 30, where $n_{\text{sample}} = 1$ corresponds to querying by single sentences. The complete experiment results can be found in Figure 8.

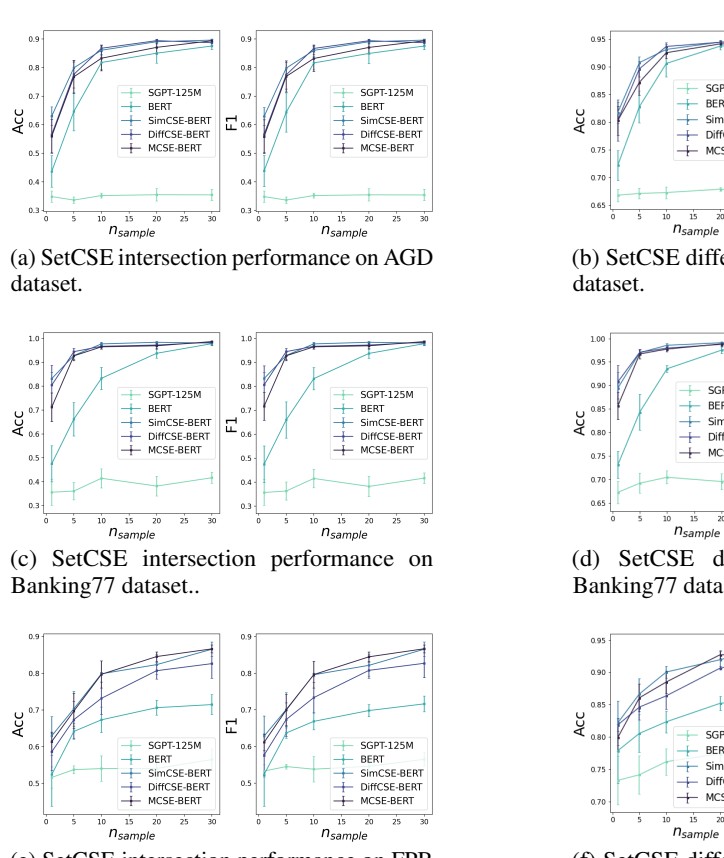

(a) SetCSE intersection performance on AGD dataset.

(b) SetCSE difference performance on AGD dataset.

(c) SetCSE intersection performance on Banking77 dataset..

(d) SetCSE difference performance on Banking77 dataset.

(e) SetCSE intersection performance on FPB dataset.

(f) SetCSE difference performance on FPB dataset.

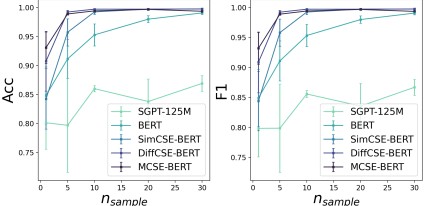

(g) SetCSE intersection performance on FMTOD dataset.

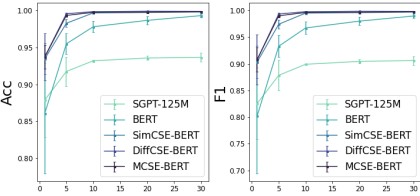

(h) SetCSE difference performance on FMTOD dataset.

Figure 8: SetCSE operation performances on AGD, Banking77, FPB, and FMTOD datasets for different values of $n_{\text{sample}}$.

### E.2 COMPARISON WITH SUPERVISED CLASSIFICATION

We also compare the SetCSE intersection performance with supervised classification, which regards the sample sentences are training sets, and predicting the class of each queried sentence. For comparing the two mechanisms, we aslo control the training epochs as the same.

The detailed results for the same evaluation datasets considered in Section 5 are listed in Table 13. As one can see, the results are on par with the one for SetCSE intersection, while supervised classification can not be used for querying semantically different sentences and conducting subsequent querying tasks.

| | AG News-T | | AG News-D | | FPB | | Banking77 | | FMTOD | |
|---|---|---|---|---|---|---|---|---|---|---|
| | Acc | F1 | Acc | F1 | Acc | F1 | Acc | F1 | Acc | F1 |
| BERT | 70.37 | 68.43 | 86.05 | 85.89 | 71.04 | 71.56 | 93.92 | 93.74 | 98.49 | 98.49 |
| RoBERTa | 75.54 | 75.56 | **88.60** | **88.64** | 75.34 | 75.58 | 83.29 | 83.29 | 99.05 | 99.05 |
| Contriever | 76.94 | 76.94 | 82.26 | 82.26 | 67.98 | 68.42 | 92.35 | 92.05 | 97.04 | 97.04 |
| SGPT | 37.55 | 37.54 | 38.21 | 38.22 | 54.99 | 55.86 | 41.59 | 41.62 | 83.77 | 84.15 |
| SimCSE-BERT | 77.77 | 77.72 | 86.14 | 86.17 | 82.52 | 82.53 | **98.63** | **98.54** | 99.31 | 99.31 |
| SimCSE-RoBERTa | **78.01** | **78.01** | 87.82 | 87.73 | **85.19** | **85.16** | 98.41 | 98.41 | 97.63 | 97.63 |
| DiffCSE-BERT | 74.11 | 74.05 | **88.49** | **88.48** | 82.61 | 82.66 | 98.45 | 98.45 | **99.66** | **99.66** |
| DiffCSE-RoBERTa | **78.37** | **78.33** | 88.29 | 88.27 | 84.36 | 84.27 | **98.53** | **98.53** | 99.14 | 99.14 |
| MCSE-BERT | 72.99 | 72.38 | 85.34 | 85.19 | 80.22 | 80.41 | 97.29 | 97.16 | **99.35** | **99.35** |
| MCSE-RoBERTa | 75.94 | 76.02 | 88.14 | 88.13 | **86.76** | **86.78** | 98.26 | 98.29 | 99.16 | 99.16 |

Table 13: Evaluation results for supervised classification ($n_{\text{sample}} = 20$).

### E.3 BENCHMARK NLU TASK PERFORMANCES POST INTER-SET CONTRASTIVE LEARNING

As mentioned in Section 5 and Appendix C, inter-set contrastive learning significantly enhances the embedding models' awareness of presented semantics. However, our interest also lies in evaluating the model's performance on general Natural Language Understanding (NLU) tasks after this context-specific fine-tuning. To this extend, we conduct evaluations on seven standard semantic textual similarity (STS) tasks (Agirre et al., 2012; 2013; 2014; 2015; 2016; Cer et al., 2017).

We first perform inter-set contrastive learning as described in the Section 5.1 experiment using the five considered datasets. Subsequently, we evaluate the model's performance on STS tasks. The model used is SimCSE-BERT, and the training hyper-parameters are the same as the ones in Section 5.1. The Spearman's correlation results for the STS tasks are presented in Table 14.

| | STS12 | STS13 | STS14 | STS15 | STS16 | STS-B |
|---|---|---|---|---|---|---|
| SimCSE-BERT | 69.03 | 77.48 | 79.21 | 83.22 | 81.74 | 83.09 |
| SimCSE-BERT (AGT) | 66.38 | 75.29 | 75.76 | 78.93 | 78.64 | **80.84** |
| SimCSE-BERT (AGD) | 64.57 | 73.18 | 74.57 | 75.58 | 74.01 | 79.14 |
| SimCSE-BERT (FPB) | 57.02 | 65.37 | 67.17 | 65.08 | 68.52 | 77.26 |
| SimCSE-BERT (Banking77) | 66.22 | **75.67** | **77.14** | **80.88** | **79.41** | 80.19 |
| SimCSE-BERT (FMTOD) | **66.39** | 72.84 | 76.82 | 80.31 | 78.95 | 80.12 |
| Ave. Change | -7% | -7% | -6% | -8% | -7% | -4% |

Table 14: Model performance on STS tasks post inter-set contrastive learning.

Notably, the application of inter-set contrastive learning exhibits no noteworthy adverse effect on the model's performance in benchmark STS tasks. On average, the utilization of inter-set contrastive learning only minimally diminishes the model's performance across STS tasks by 7% in Spearman's correlation.

