# OpenReview forum: "SetCSE: Set Operations using Contrastive Learning of Sentence Embeddings"
_ICLR.cc/2024/Conference — ICLR 2024 poster_

### Official Review · Reviewer_wrMM · 2023-10-24

**Soundness:** 2 fair
**Presentation:** 3 good
**Contribution:** 2 fair
**Rating:** 5
**Confidence:** 3

**Summary:**

This paper proposes a method to learn the sentence embedding from existing pre-trained sentence embedding to distinguish different semantics and define similarity-based set operations, including intersection and difference and their combinations. The paper evaluates the method on artificial data and shows several case studies of applications, including semantic search, data annotation, and topic discovery.

**Strengths:**

- The paper proposes a novel and simple method to fine-tune existing pre-trained embeddings to be fit to set-based operations.
- The results in the artificial setting show the feasibility of the method for several different sentence embedding
- The case studies show interesting results that present the method's potential.

**Weaknesses:**

- Representing set operation with embedding is not novel, but the comparison and discussion compared with existing methods are missing.
  - Vitalii Zhelezniak, Aleksandar Savkov, April Shen, Francesco Moramarco, Jack Flann, Nils Y. Hammerla Don't Settle for Average, Go for the Max: Fuzzy Sets and Max-Pooled Word Vectors. ICLR 2019.
  - Siddharth Bhat, Alok Debnath, Souvik Banerjee, and Manish Shrivastava. Word Embeddings as Tuples of Feature Probabilities. RepL4NLP. 2020.
  - Shib Dasgupta, Michael Boratko, Siddhartha Mishra, Shriya Atmakuri, Dhruvesh Patel, Xiang Li, and Andrew McCallum. Word2Box: Capturing Set-Theoretic Semantics of Words using Box Embeddings. ACL2022.
- The set operation presented in the paper does not satisfy the commutative law, and it orders the elements in the first set. This is not the usual set theory, and the users may be confused if they use the method that supports usual set operations, but the limitations are not discussed in detail.
- The quantitative evaluation is performed only in artificial settings, and there are only case studies for the application results. It is unclear how the method can be stably used for the application.

**Questions:**

Please see the weaknesses.

---

> ### Author Response · Authors · 2023-11-18
> **Response to Reviewer wrMM**
>
> 1. **Representing set operation with embedding is not novel, but the comparison and discussion compared with existing methods are missing.**
>
>    We would like to thank the reviewer for sharing those papers that combines word embeddings with set theory or set-theoretic operations. Our original review focused more on the works that apply set theory in word representation and comprehension, however, research that employees set theory for word embedding quality improvements are not addressed. The citations suggested by the reviewer have been incorporated into the manuscript.
>
>    Additionally, in the updated manuscript, we include a detailed discussion comparing SetCSE with these existing methods. Specifically, we aim to underscore the following as the novelty of our work:
>    * SetCSE employs **sentence embeddings** for semantic representation and information retrieval, diverging from the prior focus of the mentioned works on using and improving **word embeddings**.
>    * SetCSE utilizes sets of sentences and its learning mechanism to recognize and represent complex and intricate semantics for information querying. This approach differs from previous works, which did not consider the collective use of words to represent complex semantics.
>    * SetCSE integrates set-theoretic operations for expressing complex queries in practical sentence retrieval tasks, distinguishing it from previous works that used set operations to uncover word relationships.
>
> $ $
>
> 2. **The set operation presented in the paper does not satisfy the commutative law, and it orders the elements in the first set. This is not the usual set theory, and the users may be confused if they use the method that supports usual set operations, but the limitations are not discussed in detail.**
>
>    As per the reviewer's note, we acknowledged, in the original submission, that the SetCSE operations don't adhere to the commutative law. However, to eliminate any confusion surrounding this property, we've extensively discussed it in our updated manuscript. We've emphasized that the SetCSE operations differ from those defined in Set Theory (Cantor, 1874; Johnson-Laird, 2004). While borrowing the concept and syntax of set-theoretic operations, we've employed these to establish a querying framework. Specifically, the updated manuscript highlights the following benefits derived from this approach:
>    * It is intuitive to borrow the concepts of intersection and difference operations to describe the **selection** and **deselection** of sentences with certain semantics.
>    * Serving as a querying framework, SetCSE is designed to retrieve information from a set of sentences following certain queries. And the proposed SetCSE operation syntax aligns well with its purpose. For instance, the SetCSE serial operations $A \cap B \backslash C$ means *``finding sentences in the set A that contains the semantics B but not C.''*
>
> $ $
>
> 3. **The quantitative evaluation is performed only in artificial settings, and there are only case studies for the application results. It is unclear how the method can be stably used for the application.**
>
>    We appreciate the reviewer's suggestion. In order to better demonstrate the stability of SetCSE performance in serial operations, we have included the following experiments and evaluations:
>    * Series of two SetCSE intersection operations.
>    * Series of two SetCSE difference operations.
>    * Series of SetCSE intersection and difference operations.
>
>    The experiment are described in the Appendix C.4 of the updated manuscript. Since we are evaluating the performance of SetCSE serial operations, the multi-label datasets are employed. To encompass diverse contexts, we consider the datasets GitHub Issue (Ismael, 2022), English Quotes (Eltaief, 2022), and Reuters-21578 (Lewis, 1997). The detailed results can be found in the manuscript Tables 8 to 10 (we include one set of results below). As those results indicate, SetCSE improves the aforementioned serial operations by 26%, 39%, and 34%, respectively. In most of the cases, the accuracy and F1 in these experiments are above 70%.
>
>    **Table 8. Evaluation results for series of SetCSE intersections.**
>    |      | GitHub-HD |   Quotes-FL   | Quotes-FI  | Reuters-SG | Reuters-SC |
>    | :---        |    :----:   |   :----:   |   :----:   |   :----:   |        :---: |
>    | Metrics     | Acc  F1 | Acc  F1 | Acc  F1 | Acc  F1 | Acc  F1 |
>    | Ave. Improvement   | 55% 51% | 22% 22% | 8% 7% | 38% 34% | 13% 11% |
>
>
>    In summary, the stability of SetCSE can be evaluated through four assessments:
>    * Performance on single SetCSE operations (Section 5)
>    * Performance on series of SetCSE operations (Appendix C.4)
>    * Case studies presented in the application section (Section 6)
>    * The error analysis corresponding to the case studies (Appendix D.1)
>
>    We hope the above quantitative and qualitative results could provide more insights into the performance of SetCSE.

---

> ### Comment · Reviewer_wrMM · 2023-11-22
>
> Thank you for the response and update. Some of my concerns are addressed, so I raised my score.

---

### Official Review · Reviewer_fKmY · 2023-10-28

**Soundness:** 2 fair
**Presentation:** 3 good
**Contribution:** 3 good
**Rating:** 8
**Confidence:** 4

**Summary:**

The paper proposes a new information retrieval framework, SetCSE, based on set operation and sentence-level contrastive learning. The whole framework includes two steps: 1. fine-tuning sentence embedding model by minimizing inter-set loss. 2. ranking sentences in the set based on definition 2. The paper then defines two set operations, interactions and differences, based on the sentence-set semantic similarity. Furthermore, the paper conducts experiments for both set intersection and difference. Additionally, the paper shows three real-world applications, including semantic search, data annotation, and new topic discovery. The paper also provides quantitative analysis by comparing SetCSE with supervised learning.

**Strengths:**

1. The paper formulates the sentence retrieval problem as a combination of sentence-set similarity and set operations, which is novel for the community.
2. The paper provides a comprehensive set of experiments, introducing two new settings: set intersection and set differences. It uses multiple baselines to demonstrate the robustness of the framework. The paper also offers sentence embedding visualizations to illustrate the improvement in sentence representations. Additionally, the paper presents detailed hyperparameters and offers quantitative justification for the proposed framework
3. The paper provides three different downstream applications. Each is paired with background papers and results to show the effectiveness of the proposed methods.

**Weaknesses:**

1. The paper would benefit from an additional experiment on sentence retrieval, similar to the one in Section 6.1. In this setting, the paper could compare its performance against traditional retrieval-based methods such as DPR and BM25 to better illustrate the model's improvements. Furthermore, in Table 2, certain baseline models do not show significant improvements with SetCSE. For instance, the improvements for SGPT are marginal when compared to other baselines. The paper should include an analysis explaining the variations in improvements among different models
2. The core concept is to create clusters of sentences with semantic meaning, and this can limit the generalization ability of the proposed framework. In comparison to other baselines, the incorporation of semantic meaning within sets naturally provides additional information for training.
3. The paper fails to provide code.

**Questions:**

How to adapt the proposed method to a situation where sentences are not clustered with semantic meaning or where the clusters do not exist?

---

> ### Author Response · Authors · 2023-11-18
> **Response to Reviewer fKmY**
>
> 1. **The paper would benefit from an additional experiment on sentence retrieval, similar to the one in Section 6.1. In this setting, the paper could compare its performance against traditional retrieval-based methods such as DPR and BM25 to better illustrate the model's improvements.**
>
>     We appreciate the suggestion. In response, we have added BM25 and DPR as two baseline and compared with SetCSE. The results are included in **Tables 1** and **2** of the updated manuscript. We find that the above methods are less performant compared to SetCSE embedding models.
>
>    **Table 1. Evaluation results for intersection operation.**
>    |      | AGT |   AGD   | FPB  | Banking77 | FMTOD |
>    | :---        |    :----:   |   :----:   |   :----:   |   :----:   |          ---: |
>    | BM25     | 24.90 24.90 | 25.02 25.02 | 33.40 38.91 | 41.25 41.32 | 37.59 41.19 |
>    | DPR   | 25.00 25.00 | 25.00 25.00 | 33.33 38.81 | 41.30 41.35 | 38.33 42.87 |
>
>    **Table 2. Evaluation results for difference operation.**
>    |      | AGT |   AGD   | FPB  | Banking77 | FMTOD |
>    | :---        |    :----:   |   :----:   |   :----:   |   :----:   |          ---: |
>    | BM25     | 57.34 57.47 | 59.95 57.57 | 69.22 60.77 | 68.76 68.14 | 73.97 71.74 |
>    | DPR   | 56.50 54.45 | 56.98 54.76 | 66.67 58.06 | 68.70 68.06 | 71.67 68.43 |
>
>    $ $
>
> 2. **Furthermore, in Table 2, certain baseline models do not show significant improvements with SetCSE. For instance, the improvements for SGPT are marginal when compared to other baselines. The paper should include an analysis explaining the variations in improvements among different models.**
>
>    We have included a detailed discussion regarding the performance of SGPT embedding model and potential resolution in Appendix C.3 of our updated manuscript. Specifically, the following are addressed:
>
>    In Section 5, our evaluations demonstrate that the decoder-only SGPT-125M performs less effectively compared to encoder-based models of similar sizes, both before and after inter-set contrastive learning stages. This observation aligns with findings from other studies that compare embeddings produced by BERT-based models and GPT in benchmark word embedding tasks (Ethayarajh, 2019; Liu et al., 2020; Cai et al., 2020).
>
>    Since our evaluation indicates that SGPT benefits less from inter-set fine-tuning, future studies may consider other contrastive learning methods (Jian et al., 2022; Jain et al., 2023) to enhance the context awareness and discriminatory capabilities of decoder-based models.
>
>    $ $
>
> 3. **The core concept is to create clusters of sentences with semantic meaning, and this can limit the generalization ability of the proposed framework. In comparison to other baselines, the incorporation of semantic meaning within sets naturally provides additional information for training.**
>
>    We appreciate this comment from the reviewer. Indeed, the inter-set contrastive learning proposed in our framework aims to reinforce the semantics represented by sample sets. Hence, the learning mechanism introduced context-specific fine-tuning to the model. Evaluation presented in **Tables 1**, **2**, and **Tables 8** to **10** of the updated manuscript show that this improvements on the awareness of the query task specific semantics do enhance the query performance. In addition, the fine-tuning computation is not costly.
>
>    Furthermore, we incorporated a study in the updated manuscript to evaluate the inter-set fine-tuned models' capability to conduct general purpose NLU tasks. We conducted 7 STS tasks for the inter-set fine-tuned model, and find that the context-specific learning does not significantly impact the generalization of the underlying model, only decreasing the task performance by 7%.
>
>    Combining the fact that the fine-tuning significantly improves the querying performance, being computationally efficient, and does not create a large impact to the model's generalization capability, we think the proposed learning is worthwhile.
>
>    $ $
>
> 4. **The paper fails to provide code.**
>
>    We appreciate the interest of the reviewer regarding codes. The codes have been uploaded to the supplementary material section.
>
>    $ $
>
> 4. **How to adapt the proposed method to a situation where sentences are not clustered with semantic meaning or where the clusters do not exist?**
>
>    This potential challenge could be resolved by incorporating samples created by domain experts or generative models. For instance, if in some cases samples for certain semantics are less than say 20 or 30, and those semantics one wants to query are complex and intricate in natural, one could consult the domain expert for more samples. If the semantics are less involved, one can manually create or even query generative models for more samples. In Section 7, we also indicate that the samples needed for SetCSE to yield good results are not very large, practically, 20 to 30 samples with diversity should be sufficient.

---

> ### Comment · Reviewer_fKmY · 2023-11-22
>
> Thank you for your prompt and comprehensive reply! The authors have addressed all of my questions and did additional experiments. I raised my score to 8.

---

### Official Review · Reviewer_BQxn · 2023-10-31

**Soundness:** 4 excellent
**Presentation:** 3 good
**Contribution:** 4 excellent
**Rating:** 8
**Confidence:** 2

**Summary:**

The current research piece builds on top of existing sentence embedders based on contrastive learning by defining set operations that can be applied to sentences. Concretely, the concept of semantic similarity between sentences is extended to be defined between individual ones and Sets of sentences, by taking the average similarity.
Two new set operations are defined. The operations build upon the order relationship between the elements of the first set. The intersection of A and B is the set of elements in A that are closer to B (formally all the elements in the intersection are more similar to B than any other element not in the intersection). And the difference A minus C, corresponding to all the elements in A that are less similar to C (analogously to intersection, all the elements in the different are *less* similar to C tan the elements not in the difference).
The framework can be applied to any language models that measure sentence similarity. Experiments are carried out taking baselines as TDIDF, BERT, RoBERTa, Contriever SimCSE, DiffCSE MCSE and SGPT on the AG News, Financial PhraseBank, Banking77 and MTOD datasets.
Baselines are compared without and with the contrastive training that makes the model aware of set operations, showing an improvement in their perception of these set operations.
The work closes with a use case application. The set operations can be used to search related sentences using a set of sentences as positive or negative filtering criteria.

**Strengths:**

The idea of this paper is really clever, simple and straightforward.
The motivation is there, provide a LLM with notions of set theory to improve it in terms of search capabilities.
Experimentation seems reasonable and enough. Proves the point authors want to provid

**Weaknesses:**

The authors make a good point to show the capabilities brought by the new training regime. However, there is no analysis on capabilities that are lost because of it. Do the models train with this regime underperform on sentence similarity or information retrieval datasets.

**Questions:**

How do the models perform on sentence similarity tasks after applying the SetCSE training?

How about other general NLU tasks such as sentence classification, sequence tagging, extractive QA or multiple choice QA?

The largest model where this approach was applied was a RoBERTa-like model. Does the approach escalate to bigger models?

Can it be applied while using LoRA?

---

> ### Author Response · Authors · 2023-11-18
> **Response to Reviwer BQxn**
>
> 1. **The authors make a good point to show the capabilities brought by the new training regime. However, there is no analysis on capabilities that are lost because of it. Do the models train with this regime underperform on sentence similarity or information retrieval datasets. How do the models perform on sentence similarity tasks after applying the SetCSE training? How about other general NLU tasks such as sentence classification, sequence tagging, extractive QA or multiple choice QA?**
>
>    We're thankful for the reviewer's suggestions. As previously indicated in the original manuscript, one of our intended explorations involves assessing the models' generalization capability on benchmark NLU tasks subsequent to inter-set contrastive learning. In addressing the reviewer's comments, we proceeded with evaluations on 7 standard benchmark Semantic Textual Similarity (STS) tasks (Agirre et al., 2012, 2013, 2014, 2015, 2016; Cer et al., 2017).
>
>    We first perform inter-set contrastive learning as described in the Section 5.1 experiment using the five datasets - AGT, AGD, FPB, Banking77, and FMTOD. Subsequently, we evaluate the model’s performance on STS tasks. The model used is SimCSE-BERT, and the training hyper-parameters are the same as the ones in Section 5.1. The Spearman’s correlation results for the STS tasks are presented in **Table 14**.
>
>    **Table 14: Model performance on STS tasks post inter-set contrastive learning.**
>    |      | STS12 |   STS13   | STS14  | STS15 | STS16 | STSB |
>    | :---        |    :----:   |   :----:   |   :----:   |   :----:   |    :----:     |   :---:    |
>    | SimCSE-BERT | 69.03 | 77.48 | 79.21 | 83.22 | 81.74 | 83.09 |
>    |SimCSE-BERT (AGT) |  66.38 | 75.29 | 75.76 | 78.93 | 78.64 | 80.84|
>    |SimCSE-BERT (AGD) | 64.57 | 73.18 | 74.57 | 75.58 | 74.01 | 79.14 |
>    |SimCSE-BERT (FPB) | 57.02 | 65.37 | 67.17 | 65.08 | 68.52 | 77.26 |
>    |SimCSE-BERT (Banking77) | 66.22 | 75.67 | 77.14 | 80.88 | 79.41 | 80.19 |
>    |SimCSE-BERT (FMTOD) | 66.39 | 72.84 | 76.82 | 80.31 | 78.95 | 80.12 |
>    |Ave. Change | -7% | -7% | -6% | -8% | -7% | -4% |
>
>    Notably, the application of inter-set contrastive learning exhibits no noteworthy adverse effect on the model’s performance in benchmark STS tasks. On average, the utilization of inter-set contrastive learning only minimally diminishes the model’s performance across STS tasks by 7% in Spearman’s correlation.
>
>    $ $
>
> 2. **The largest model where this approach was applied was a RoBERTa-like model. Does the approach escalate to bigger models? Can it be applied while using LoRA?**
>
>    We value the reviewer's suggestion to apply SetCSE to larger embedding models, aiming for enhanced performance. Presently, we recognize two main types of embedding models suitable for our framework: encoder-based Transformer models like BERT, RoBERTa, and their fine-tuned versions, as well as decoder-based embedding models like GPT. Among the former, BERT and RoBERTa-based models continue to achieve state-of-the-art results in specific NLU tasks, such as SMART-RoBERTa (Jiang et al., 2019) and StructBERT (Wang et al., 2020) in STS benchmark tasks. To further enhance larger models with SetCSE and achieving better results, additional work can be focused on decoder-based models, some of which have parameter counts reaching hundreds of billions. In the updated manuscript, we've included a thorough discussion in Section 5.1 and Appendix C.3, proposing adjustments to the inter-set contrastive learning mechanism to better suit decoder-based models.
>
>    Additionally, we have outlined plans for future work to incorporate LoRA into SetCSE in Section 8.

---

### Official Review · Reviewer_3g7o · 2023-11-03

**Soundness:** 3 good
**Presentation:** 3 good
**Contribution:** 3 good
**Rating:** 6
**Confidence:** 4

**Summary:**

This paper introduces Set Operations using Contrastive Learning of Sentence Embeddings (SetCSE), a framework that incorporates set theory into semantic search to handle complex and multifaceted queries. Recognizing that intricate semantics often arise from clusters of sentences rather than isolated ones, the authors present a method where sets of sentences collectively represent semantics. The inter-set contrastive learning component is designed to fine-tune language models to capture contextual nuances and discern between different semantic sets. The SetCSE operations - intersection, difference, and series - are employed to structure queries effectively, allowing for a granular and nuanced retrieval of sentences from large corpora. An illustrative use case of analyzing S&P 500 companies’ stances on ESG issues demonstrates the framework's practical utility in complex information retrieval scenarios.

**Strengths:**

1. Provides a well-defined and practical framework that enables complex information retrieval tasks which are not possible with current search methodologies.

2. Applies contrastive learning to sentence embeddings in a unique way, emphasizing contextual differentiation between sets of sentences.

3. Offers compelling real-world applications, such as parsing nuanced topics like ESG stances from earnings calls, highlighting the framework’s potential for practical deployment.

**Weaknesses:**

1. There is no mention of an error analysis which would be beneficial in understanding the limitations of SetCSE in certain scenarios.

2. The applications of complex semantic search, data annotation, and new topic discovery are very cool with the detailed examples, but there is not quantification here or comparison with others with existing set methods from the literature (same with Table 1 and 2 as well). Do you have comparisons with other methods from the literature on this topic?

Typos:
Section 7 "DISUCSSION"

**Questions:**

1. Can the authors discuss any observed limitations or frequent error patterns during SetCSE operations?

2. How does SetCSE scale with the size of the dataset and the complexity of the query semantics?

---

> ### Author Response · Authors · 2023-11-18
> **Response to Reviewer 3g7o - Part 1**
>
> 1. **There is no mention of an error analysis which would be beneficial in understanding the limitations of SetCSE in certain scenarios.**
>
>    We appreciate reviewer's constructive suggestion. In response, we have include the error analysis the error analysis with respect to Section 6.1 use cases in **Table 11** of the updated manuscript. Specifically, **Tables 11a** and **11b present** the less preferable query results compared to **Tables 3b** and **3c**, respectively. It’s important to note that the presented results are not ranked among the top sentences in the corresponding SetCSE querying outputs. Instead, they are listed between the 30th to 50th positions within the sentences of the S&P500 earnings call dataset (Qin & Yang, 2019). See below as a highlight.
>
>    **Table 11a. Error analysis for the query ''*using technology to solve Social issues, while neglecting its potential negative impact*".**
>    | |
>    | :---------- |
>    | Finally, there is one commonality our **customers** have, it’s that they live in a hybrid **IT world**. |
>    | We had good **bio-security** to begin with, but we amped it up to an all-time high level of **discipline and scrutiny**, frankly, and it hasn’t stopped.|
>    | It is fairly incremental in terms of adding things like **customer support**, **field application engineering, software support**, given that we’re familiarizing people with our architecture.|
>    | |
>
>    Based on the findings in **Table 11**, we identify the following issues that may lead to less preferred results:
>    * Certain words in the query results closely match sample sentences in terms of represented semantics, leading to their respective rankings. However, the entire sentences show less relevance to those specific semantics. For instance, word “*customer*” may align with the sample phrase “*customer satisfaction*” yet the entire sentence might be less associated with “*Social issues*”.
>
>    * Some sentences are chosen due to their high relevance to a single presented semantic, while an intersection of multiple semantics is anticipated. For example, the sentence “*We had good **bio-security** to begin with, but we amped it up to an **all-time high level of discipline and scrutiny**, frankly, and it hasn’t stopped.* ” aligns closely with “*Social issue*” but displays less relevance to “new technology”.
>
>    It’s essential to note that the mentioned examples are not ranked among the top sentences in the query outputs, thereby naturally leading to errors. We aim to improve the assessment of the closeness between a sentence and sets of semantics to mitigate some of these aforementioned errors.
>
>    $ $
>
> 2. **The applications of complex semantic search, data annotation, and new topic discovery are very cool with the detailed examples, but there is not quantification here or comparison with others with existing set methods from the literature (same with Table 1 and 2 as well). Do you have comparisons with other methods from the literature on this topic?**
>
>    We would like to thank the reviewer for the two suggestions: (1). further quantifying SetCSE performance, (2). compare SetCSE with more works presented in the literature.
>
>    **In response of the first suggestion**, an extensive evaluation on SetCSE serial operations performance is presented in Appendix C.4, which considers the following three SetCSE serial operations:
>    * two SetCSE intersection operations.
>    * two SetCSE difference operations.
>    * SetCSE intersection and difference operations.
>
>    We employ the multi-label datasets for serial operation evaluation. To encompass diverse contexts, we consider GitHub Issue (Ismael, 2022), English Quotes (Eltaief, 2022), and Reuters-21578 (Lewis, 1997) datasets. Detailed results can be found in the Tables 8-10 of the updated manuscript. As an example, the results indicate an average improvement of 34% in the performance of serial operations using SetCSE.
>
>    **Table 8. Evaluation results for series of two SetCSE intersections.**
>    |      | GitHub-HD |   Quotes-FL   | Quotes-FI  | Reuters-SG | Reuters-SC |
>    | :---        |    :----:   |   :----:   |   :----:   |   :----:   |        :---: |
>    | Metrics     | Acc  F1 | Acc  F1 | Acc  F1 | Acc  F1 | Acc  F1 |
>    | Ave. Improvement   | 55% 51% | 22% 22% | 8% 7% | 38% 34% | 13% 11% |
>
>    **In response to the second suggestion**, we included the following to the manuscript:
>
>    * More related works that combines Set Theory with embeddings (Section 2). We indicate that these works only focus on at word level, with a end-goal of improving word embeddings or revealing word relationships, where SetCSE operates on sentence level, with a purpose of building a robust and practical querying framework for complex semantics.
>    * A quantitative evaluation for the existing information retrieval methods (Section 5, Tables 1 and 2), including BM25 and DPR. We found that SetCSE performance is more than two times compared with these methods in terms of accuracy and F1.

---

> ### Author Response · Authors · 2023-11-18
> **Response to Reviewer 3g7o - Part 2**
>
> 3. **Can the authors discuss any observed limitations or frequent error patterns during SetCSE operations?**
>
>    Please see response 2 for error analysis. In terms of other observed limitations, we want to point out that the SetCSE querying framework depends on the presented samples to improve the model discriminatory capability over complex semantics. This context-specific method is not designed for general NLU benchmark tasks, and we do observe a small decrease in model performance in standard STS tasks (see **Table 14** of the updated manuscript). However, this would not affect the usage of SetCSE as a querying framework, as we see that the inter-set contrastive learning significantly improves model performance in information retrieval when complex semantics and query structures are involved.
>
>    $ $
>
> 4. **How does SetCSE scale with the size of the dataset and the complexity of the query semantics?**
>
>    We thank the reviewer for the question of SetCSE scalability. To illustrate, we include the following table regarding the five dataset used in our quantitative evaluations in Section 5 for more insights:
>
>    **Table. Statistics for dataset AGT, AGD, FPB, Banking77, and FMTOD.**
>
>    |              | AGT |   AGD   | FPB  | Banking77 | FMTOD |
>    | :---        |    :----:   |   :----:   |   :----:   |   :----:   |   :---:   |
>    | # of sentences     | 12,010 | 12,010 | 5,717 | 10,003 | 33,698 |
>    | Ave. sentence length  | 41.97 | 193.55 | 122.71 | 59.47 | 35.70 |
>
>    From the above table, we find that the AGD and FPB datasets are equipped with longer sentences, while FMTOD dataset contains most sentences among other datasets. In addition, the semantics presented in AGT, AGD and FPB are more general hence harder to "comprehend" by the model, whereas semantics in Banking77 and FMTOD are more specific, and easy to identify based on the key words presented. Based on the above, and the results presented in **Table 1** and **2** in the manuscript, we conclude that SetCSE performance are less impacted by the size of the dataset, while it generally performances better when semantics are specific instead of general.
>
>    $ $
>
> 5. **Typos: Section 7 "DISUCSSION"**
>
>    The typo has been corrected. We appreciate the reviewer's help.

---

### Author Response · Authors · 2023-11-18
**Response to Reviewers**

We would like to thank all the reviewers for their insightful and constructive feedback. It's rewarding to see this work acknowledged for its novelty, practicality, straightforwardness, and potential within the domain.

The constructive feedback provided by the reviewers has significantly contributed to enhancing the quality of our paper. In response to this feedback, we have introduced updates to our manuscript, which are indicated in blue for easy identification. We've addressed each reviewer's comments individually and also highlighted the major changes as follows.

**1**. Incorporating three new sets of experiments to quantitatively assess the performance and stability of SetCSE series of operations. Specifically, these experiments cover the following three scenarios:
   * Series of SetCSE intersection operations
   * Series of SetCSE difference operations
   * Series of SetCSE intersection and difference operations

   Since we are evaluating the performance of SetCSE serial operations, the multi-label datasets are employed. To encompass diverse contexts, we consider the datasets GitHub Issue (Ismael, 2022), English Quotes (Eltaief, 2022), and Reuters-21578 (Lewis, 1997). The results indicate an average improvement of 34% in the performance of serial operations using SetCSE. Also see below for the average improvements obtained for the three aforementioned experiments.

   **Table 8. Evaluation results for series of two SetCSE intersections.**
   |      | GitHub-HD |   Quotes-FL   | Quotes-FI  | Reuters-SG | Reuters-SC |
   | :---        |    :----:   |   :----:   |   :----:   |   :----:   |        :---: |
   | Metrics     | Acc  F1 | Acc  F1 | Acc  F1 | Acc  F1 | Acc  F1 |
   | Improvement   | 55% 51% | 22% 22% | 8% 7% | 38% 34% | 13% 11% |

   **Table 9. Evaluation results for series of two SetCSE difference operations.**
   |      | GitHub-HD |   Quotes-FL   | Quotes-FI  | Reuters-SG | Reuters-SC |
   | :---        |    :----:   |   :----:   |   :----:   |   :----:   |        :---: |
   | Metrics     | Acc  F1 | Acc  F1 | Acc  F1 | Acc  F1 | Acc  F1 |
   | Improvement   | 47% 45% | 15% 12% | 32% 29% | 55% 52% | 53% 48% |

   **Table 10. Evaluation results for series of SetCSE intersection and difference.**
   |      | GitHub-HD |   Quotes-FL   | Quotes-FI  | Reuters-SG | Reuters-SC |
   | :---        |    :----:   |   :----:   |   :----:   |   :----:   |        :---: |
   | Metrics     | Acc  F1 | Acc  F1 | Acc  F1 | Acc  F1 | Acc  F1 |
   | Improvement   | 27% 22% | 24% 21% | 41% 39% | 57% 54% | 41% 37% |

   $ $

**2**. Introducing an error analysis for the use cases presented in Section 6 to identify patterns in less preferred results from SetCSE queries. The updated manuscript thoroughly discusses error patterns and potential resolutions.

   $ $

**3**. Adding evaluations for traditional retrieval-based techniques like BM25 and DPR in our experiments, listed below, and providing more insights into the advantages offered by SetCSE in comparison to existing methods.

   **Table 1. Evaluation results for intersection operation.**
   |      | AGT |   AGD   | FPB  | Banking77 | FMTOD |
   | :---        |    :----:   |   :----:   |   :----:   |   :----:   |          ---: |
   | BM25     | 24.90 24.90 | 25.02 25.02 | 33.40 38.91 | 41.25 41.32 | 37.59 41.19 |
   | DPR   | 25.00 25.00 | 25.00 25.00 | 33.33 38.81 | 41.30 41.35 | 38.33 42.87 |

   **Table 2. Evaluation results for difference operation.**
   |      | AGT |   AGD   | FPB  | Banking77 | FMTOD |
   | :---        |    :----:   |   :----:   |   :----:   |   :----:   |          ---: |
   | BM25     | 57.34 57.47 | 59.95 57.57 | 69.22 60.77 | 68.76 68.14 | 73.97 71.74 |
   | DPR   | 56.50 54.45 | 56.98 54.76 | 66.67 58.06 | 68.70 68.06 | 71.67 68.43 |

   $ $

**4**. Evaluating the impact of inter-set contrastive learning on the generalization capability of embedding models. This evaluation involves assessing the fine-tuned embedding models' performance on benchmark NLU tasks, including 7 semantic textual similarity (STS) tasks, demonstrating no significant adverse effect on their performance. See below.

   **Table 14. Model performance on STS benchmark post inter-set contrastive learning.**
   |      | STS12 |   STS13   | STS14  | STS15 | STS16 | STSB |
   | :---        |    :----:   |   :----:   |   :----:   |   :----:   |    :----:     |      ---: |
   | post fine-tuning SimCSE-BERT (Ave.)    | 64.12 | 72.37 | 74.29 | 76.16 | 75.91 | 79.51|

   $ $

**5**. Adding more citations regarding the combination of set theory and word representations. The commonality of those works comes from their employment word embeddings. The novelty of SetCSE compared with the cited papers are discussed in the updated manuscript as well.

   $ $

**6**. Discussing in greater detail the advantages and limitations of the SetCSE operations definition.

   $ $

**7**. Introducing a detailed discussion regarding the performance of the decoder-based SGPT model.

---

### Meta-Review · Area_Chair_6HNx · 2023-12-13

**Metareview:**

The paper introduces an information retrieval framework that accounts for sentence cluster semantic representation.  The authors seem to respond (via extra experiments, explanations and analysis) to all reviewers.

**Justification For Why Not Higher Score:**

The paper seems to lie on the positive side of a borderline paper.

**Justification For Why Not Lower Score:**

Generally positive reviews.  Sound paper.

---

### Decision · Program_Chairs · 2024-01-16

Accept (poster)